# Eye activity tracks task-relevant structures during speech and auditory sequence perception

Peiqing Jin[1], Jiajie Zou[1], Tao Zhou[1] & Nai Ding[1,2,3]

The sensory and motor systems jointly contribute to complex behaviors, but whether motor systems are involved in high-order perceptual tasks such as speech and auditory comprehension remain debated. Here, we show that ocular muscle activity is synchronized to mentally constructed sentences during speech listening, in the absence of any sentence-related visual or prosodic cue. Ocular tracking of sentences is observed in the vertical electrooculogram (EOG), whether the eyes are open or closed, and in eye blinks measured by eyetracking. Critically, the phase of sentence-tracking ocular activity is strongly modulated by temporal attention, i.e., which word in a sentence is attended. Ocular activity also tracks high-level structures in non-linguistic auditory and visual sequences, and captures rapid fluctuations in temporal attention. Ocular tracking of non-visual rhythms possibly reflects global neural entrainment to task-relevant temporal structures across sensory and motor areas, which could serve to implement temporal attention and coordinate cortical networks.

[1] Key Laboratory for Biomedical Engineering of Ministry of Education, Interdisciplinary Institute of Neuroscience and Technology, Qiushi Academy for Advanced Studies, College of Biomedical Engineering and Instrument Sciences, Zhejiang University, 310027 Hangzhou, China. [2] State Key Laboratory of Industrial Control Technology, Zhejiang University, Hangzhou 310027, China. [3] Interdisciplinary Center for Social Sciences, Zhejiang University, 310027 Hangzhou, China. These authors contributed equally: Jiajie Zou, Tao Zhou. Correspondence and requests for materials should be addressed to N.D. (email: ding_nai@zju.edu.cn)

The brain is a complex system and sensorimotor interactions are involved even for seemingly simple behavior. For example, when humans or animals hear an unexpected noise, a sequence of sensorimotor processes could be triggered, e.g., increasing arousal, attending to the sound, looking in the direction where the sound comes from, and preparing for further responses, such as escaping or approaching the sound source[1,2]. In other words, a burst of noise may trigger not only auditory processing but also a sequence of movements to actively acquire further multisensory information and take action. The importance of sensorimotor integration is even more evident during complex behaviors involving speech or music: For example, human listeners can precisely tap or dance to the rhythm they hear[3]. Sensorimotor processes are also required to support turn taking during conversations[4,5] and coordination with other performers during ensemble music performance[6].

Even when no overt movement is involved, it has also been proposed that sensorimotor mechanisms play critical roles in speech and auditory perception. It has been hypothesized that the motor cortex contributes to decoding phonetic information in speech[7,8]. Neurophysiological evidence consistent with this hypothesis has shown that neural activity can track acoustic features not only in auditory cortex but also in broad frontal/parietal areas that overlap with the motor and attention networks[9–13]. Furthermore, transcranial magnetic stimulation (TMS) of the motor cortex can alter how the brain processes auditory syllables or words[14,15], and during TMS speech processing can modulate peripheral tongue/lip muscle activity[16,17]. Speech perception and auditory perception in general, however, involve units on multiple time scales, including both local units, e.g., syllables in speech and notes in music, and high-level structures, such as phrases in speech and music[18]. Recent magnetoencephalogram (MEG) and electroencephalogram (EEG) studies have demonstrated neural tracking of high-level speech and music structures[19,20], on top of neural tracking of basic sound units. It is less clear, however, whether processing high-level structures in speech and other sounds engages the motor system and whether motor activity, either cortical or muscle activity, spontaneously tracks such larger structures without any movement-related task.

Processing large structures in speech and other complex sound sequences is a challenging task, since individual sound elements come rapidly (i.e., ~4–5 syllables per second for speech[21] and ~1–4 beats per second for music[22]) and long-distance dependency exists across seconds[23,24]. The dual requirements of high processing speed and long temporal integration window force the brain to develop strategies to preferentially process words that are more informative or less predictable[25–27]. Studies on sensory processing have proposed that selective information processing in time, i.e., temporal attention, is implemented by low-frequency neural oscillations in the sensorimotor system[28–30], and can be facilitated by overt movements[2,31–33]. Neurophysiological evidence supporting this hypothesis mostly comes from studies on complex scenes consisting of multiple sensory sequences, which show that cortical activity is preferentially synchronized to the attended sensory sequence[34,35]. Within a single sequence, e.g., a speech stream, it remains to be established whether the phase of sensorimotor activity is locked to the units that are preferentially processed, and whether such activity can modulate muscle activity.

Here, we investigate whether sensorimotor activity can track high-level structures in sensory sequences, including speech and non-linguistic auditory/visual sequences, and whether it reflects temporal attention. We record from both the brain and the peripheral oculomotor system. The eyes do not belong to the classic speech/auditory pathway, but ocular muscle activity can be controlled by cortex[36,37] and can therefore reflect cortical motor activity. More importantly, it is well established that eye activity, including pupil dilation, eye movements, and blinks, is sensitive to attention and cognitive load[38–40]. Using a series of experiment involving speech, non-speech sound sequences, and visual sequences, we show that eye activity is synchronized to high-level structures in sensory sequences and is modulated by temporal attention. These results provide strong evidence that the motor/attention networks are engaged during speech/auditory perception. It also strongly suggests that the rhythms of high-level structures may serve as a synchronization signal to coordinate neural processing across massive cortical networks during sequence processing.

## Results

**EEG tracking of spoken sentences.** In a series of experiments, participants listen to a sequence of syllables that are acoustically independent but linguistically organized into higher-order structures, i.e., phrases and sentences (Fig. 1a). In the first experiment, the listeners press keys to indicate whether the last sentence they hear in a trial is complete, i.e., containing both a subject and a predicate. They closed their eyes during the experiment. EEG and electrooculogram (EOG) are simultaneously recorded. The EEG response from the listeners shows three peaks in the spectrum, at the sentential, phrasal, and syllabic rates, respectively (Fig. 1b), consistent with previous studies[20]. The EEG response topography shows a central distribution at all three peak frequencies (Fig. 1c). In Fig. 1 and other figures, the EEG spectrum is shown for channel Cz, since it contains strong responses to speech[41] and is less likely to be contaminated by eye activity compared with frontal channels. Since only the syllabic-rate rhythm is present in the stimulus, sentential-rate and phrasal-rate responses can only reflect neural tracking of mentally constructed linguistic structures.

**EOG tracking of spoken sentences.** On top of EEG, the vertical EOG also shows a significant sentential-rate response in the spectrum (Fig. 1b, Supplementary Table 1), even if the eyes are closed during the experiment and no speech-relevant visual information is presented. Importantly, the sentence-rate EOG power is not significantly correlated with the sentence-rate EEG power across participants (Fig. 1d), suggesting that EOG and EEG capture separate response components. Furthermore, within individual participants, the sentence-rate EEG power is not correlated the sentence-rate EOG power across trials (Supplementary Fig. 1).

**Eye blinks track spoken sentences.** In the second experiment, eyetracking is employed to investigate which kind of eye activity, e.g., blinks, saccades, or changes in pupil size, contributes to the sentence-tracking EOG response. The eyetracker monitors eye activity optically and electrophysiological signals, including EOG and EEG, are simultaneously recorded. The first condition is a free viewing condition, in which the listeners watch dots randomly moving on a screen. Movement of the dots is independent of concurrently presented speech. Eyetracking data, including the pupil size and binary time series denoting the occurrence of blinks and saccades, are epoched based on the onset of speech stimuli and averaged over epochs. Eyetracking data show that eye blinks show clear sentential-rate fluctuations, while the pupil size is not strongly synchronized to speech (Fig. 2a(i)). The saccade rate also shows a weak response to sentences. Furthermore, the 1-Hz blink signal and the 1-Hz vertical EOG are strongly correlated across participants (Fig. 2b), demonstrating that the 1-Hz vertical EOG response mainly reflects eye blinks during free viewing of random dots.

The next condition is also a free viewing condition but the participants look at a blank screen in a dark room. This condition presents no visual stimulus but eye blinks still show sentential-rate fluctuations (Fig. 2a(ii)). The third condition has the same setup as the second condition but the participants have their eyes closed. Since the eyes are closed, the pupil could not be detected

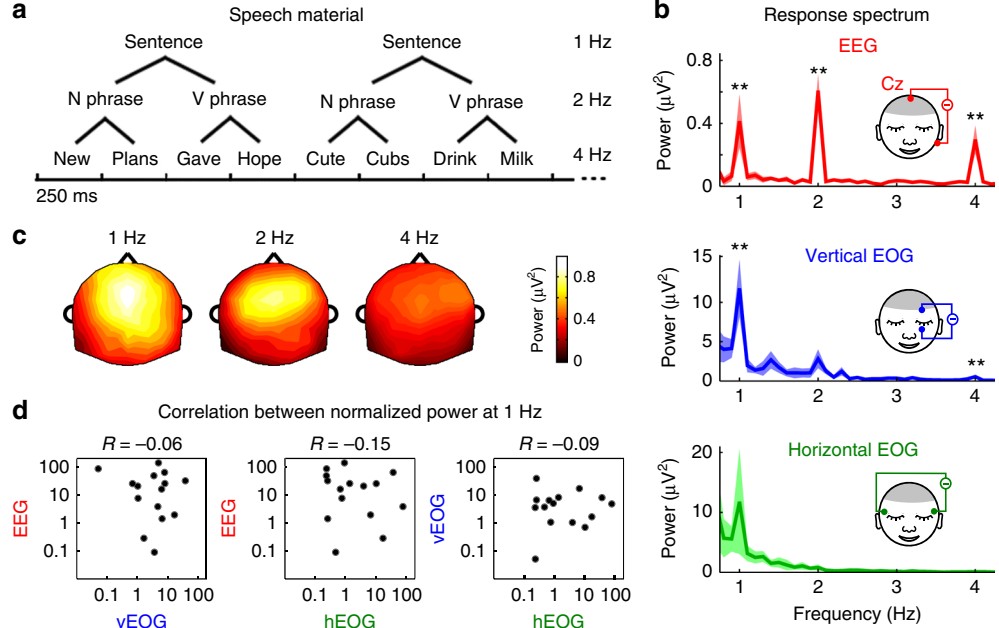

**Fig. 1** Tracking of spoken sentences in EEG and EOG. **a** The speech material consists of a sequence of four-word sentences (monosyllabic words), which can be decomposed into a two-syllable noun phrase followed by a two-syllable verb phrase. The syllables are presented at a constant rate of 4 Hz, and therefore the presentation rates of phrases and sentences are 2 and 1 Hz, respectively. English words are shown in the illustration but the actual speech materials are in Chinese. **b** The EEG spectrum (channel Cz) shows three peaks of similar magnitude at the sentential, phrasal, and syllable rates (1, 2, and 4 Hz, respectively). The vertical EOG shows a clear sentential-rate response and a very weak syllabic-rate response. The horizontal EOG also shows a sentential-rate response peak but this response shows large individual variability and does not reach statistical significance. The significance of the spectral peaks is evaluated using bootstrap (the *P*-value, mean, and SEM reported in Supplementary Table 1). The shaded area covers 1 SEM across participants (*N* = 15) on each side, evaluated by bootstrap (same for plots of spectrum/waveform in other figures). In the illustration, the reference electrode is placed on one side but in the analysis the signal is referenced based on the average mastoid recording from both sides. **c** The topography of the EEG response shows a fronto-central distribution. **d** Sentential-rate response power is not correlated across the three measures, i.e., EEG, vertical EOG, and horizontal EOG (correlation not significantly larger than 0, *P* = 0.65, 0.75, and 0.73, respectively, one-sided bootstrap, not corrected for multiple comparisons). The power at each frequency is normalized by dividing the power averaged over two neighboring frequency bins. Each black dot is the data from one individual. *$P <$ 0.05; **$P <$ 0.01

optically. The EOG recordings, however, are not affected by the closed eyes. In this condition, both EOG and EEG signals still show clear sentential-rate fluctuations (Fig. 2a(iii)). In an additional control condition, a sequence of syllables is presented in a random order, not constructing phrases or sentences. In this condition the 1-Hz response is not significantly stronger than the response averaged over neighboring frequency bins for EEG, vertical EOG, horizontal EOG, and blinks (Fig. 2a(iv)), confirming that the sentential structure is necessary to generate a 1-Hz response.

**Attention modulates ocular and neural tracking of speech.** Experiments 1 and 2 establish that eye activity can track the rhythm of spoken sentences but the underlying mechanisms remain unclear. One candidate mechanism is temporal attention. There is a well-established relationship between eye activity and visual attention[42] and it is possible that attention to speech also modulates eye activity. To test if temporal attention plays a role in driving ocular synchronization to sequence structures, a third experiment was carried out which manipulates the attentional focus within a sentence (Fig. 3a and Supplementary Fig. 2). Listeners are asked to attend to the 1st syllable of every sentence in one condition and the 3rd syllable in the other condition. The task is to detect if a cued consonant appears in the attended syllables. These two conditions employ the same speech materials and only differ in the listeners' attentional focus. If the response phase is locked to the attentional focus, i.e., the attended syllable,

there will be a ~180° phase difference between conditions, i.e., attending to the 1st or the 3rd syllable in a four-syllable sentence.

The waveforms of the EOG and EEG show rhythmic fluctuations and are strongly modulated by the listeners' attentional focus (Fig. 3b). Attentional modulation is especially apparent when the EOG/EEG responses are filtered into a narrow frequency band centered at 1 Hz. The power spectrum shows spectral peaks at 1 and 2 Hz for both EOG and EEG, in both conditions (Fig. 3c). To quantify the phase difference between the responses, we applied the Fourier transform to the response waveforms in Fig. 3b and extracted the response phase from the Fourier coefficient (Fig. 3d). At 1 Hz, the phase difference between conditions is 71° (99%/95% confidence interval: 38–105°/45–96°), 142° (74–234°/94–205°), and 172° (59–243°/107–219°), for EEG, vertical EOG, and horizontal EOG, respectively. The 99% confidence interval for phase difference deviates 0° for all three measures. Therefore, at 1 Hz, both the EOG and EEG phases are modulated by temporal attention but the EOG phase is more strongly modulated. At 2 Hz, the phase difference between conditions is −10° (−23° to 1°/−19° to −1°), −9° (−54° to 62°/−43° to 35°), and 134° (9–263°/40–241°), for EEG, vertical EOG, and horizontal EOG, respectively. The 99% confidence interval for phase difference only deviates 0° for horizontal EOG. Therefore, for EEG and vertical EOG, attention selectively modulates the sentential-rate neural response. For horizontal EOG, no significant 2-Hz peak is observed in Experiments 1 and 2, and therefore the very weak 2-Hz response

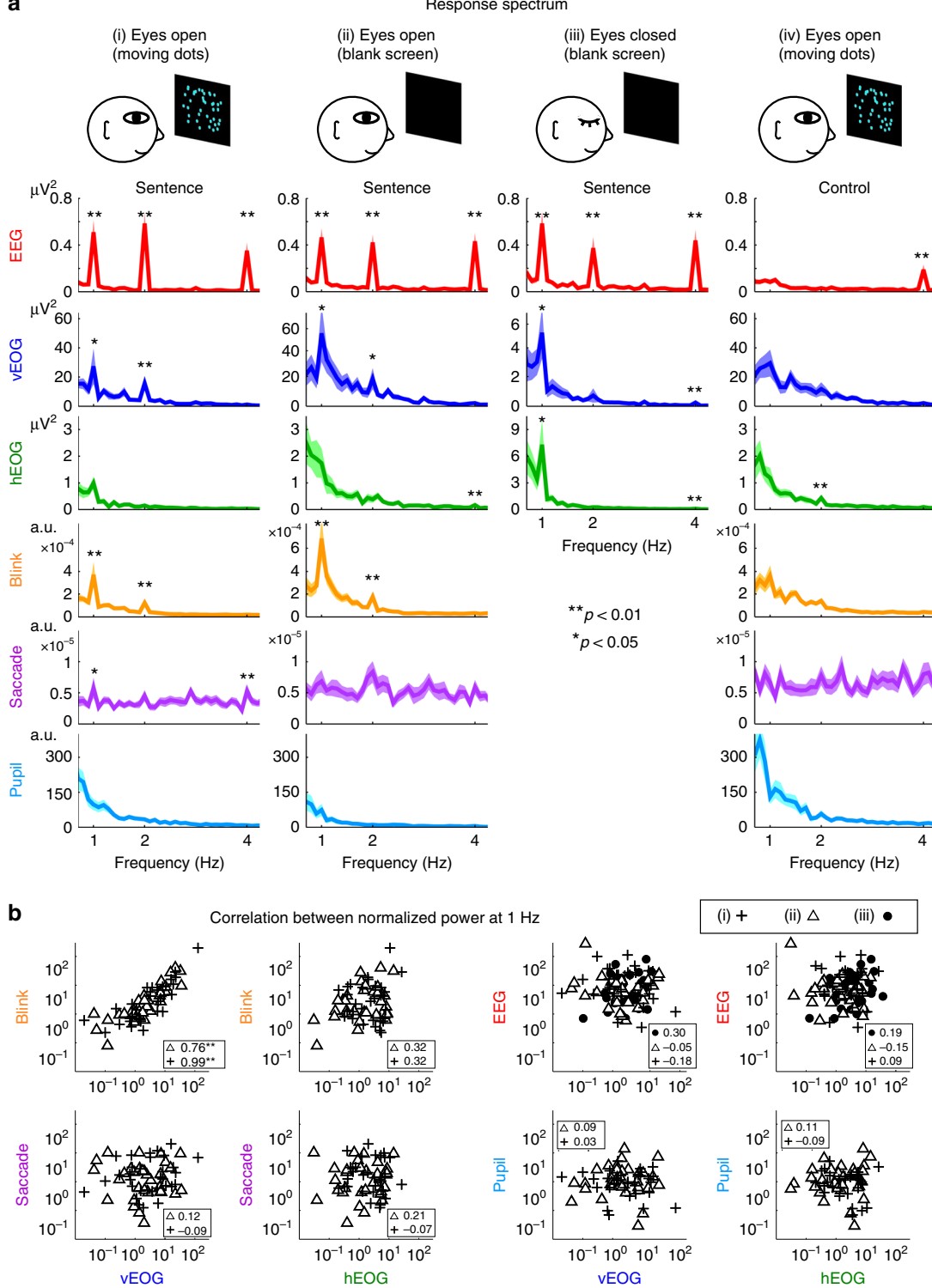

**Fig. 2** EEG, EOG, and eyetracking responses to sentences or random syllables. **a** Response spectra in four experimental conditions (columns) and for six neural/ocular measures (rows). The control condition presents random syllables instead of sentences. The blink rate measured by eyetracking, but not the saccade rate or pupil size, shows a reliable spectral peak at 1 Hz across conditions. The EEG signal is from channel Cz. Significance of each spectral peak is evaluated using bootstrap. The shaded area covers 1 SEM across participants ($N = 32$) on each side. **b** The 1-Hz blink response is correlated with the 1-Hz vertical EOG response. In contrast, the 1-Hz components of other measures are not significantly correlated. Each dot is one participant. For all significance tests in the figure, the $P$-value, mean, and SEM are reported in Supplementary Table 2

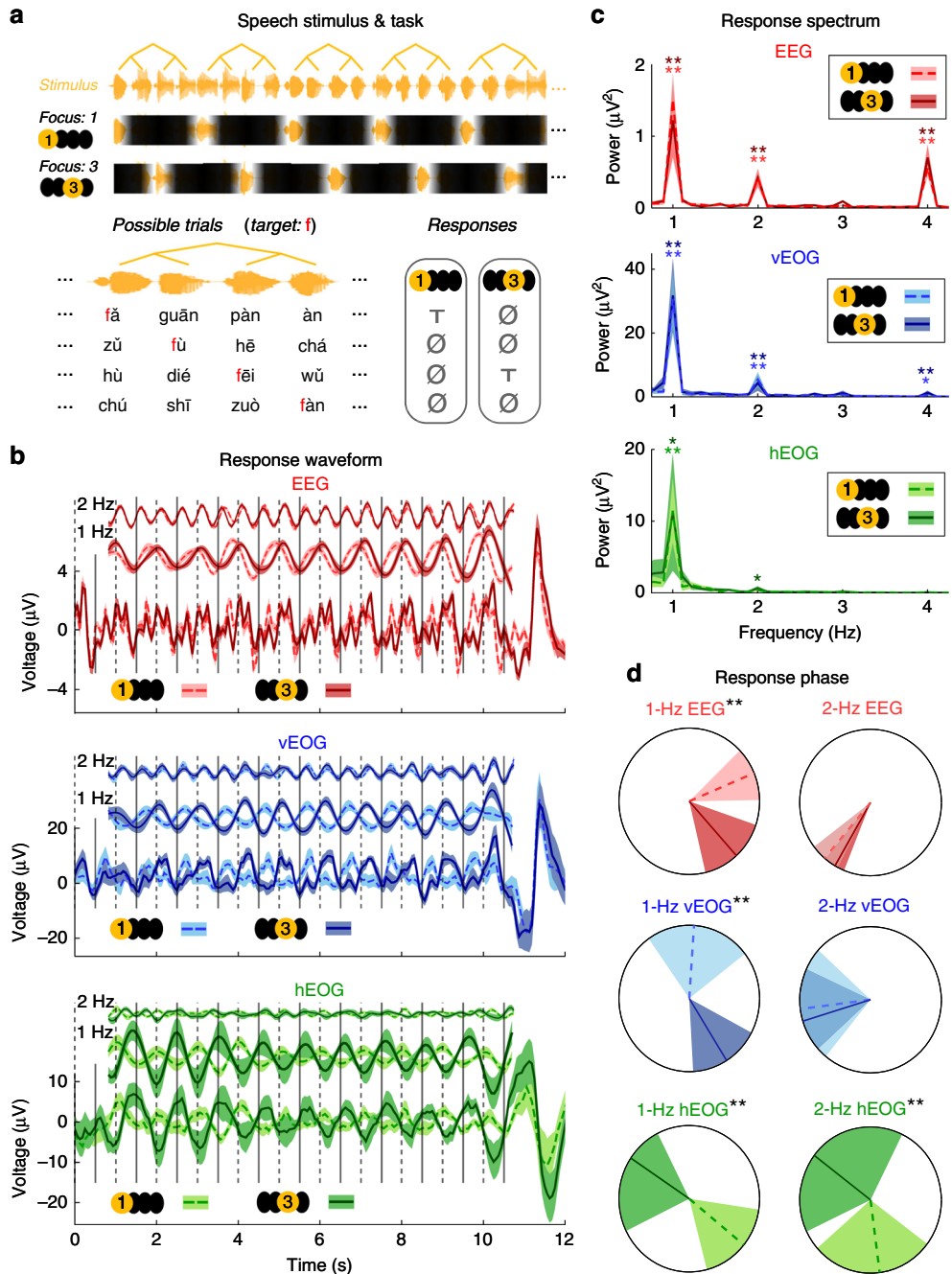

**Fig. 3** Temporal attention modulates the phase of sentence-tracking EEG/EOG activity. **a** Stimulus and task. The stimulus is a sequence of four-syllable sentences (shown in orange). The listeners have to attend to the 1st syllable of every sentence in one condition and attend to the 3rd syllable instead in the other condition. Unattended syllables are shaded in the illustration. The target phoneme, e.g., /f/, appears once in every trial but it may appear as the 1st, 2nd, 3rd, or 4th syllable in a sentence with equal probability. The listeners respond by pressing a key only when the target phoneme appears in the syllables they attend to. In the illustration, T means that a target appears while Ø means that no target appears. The trial structure is described in more details in Supplementary Fig. 2. **b** Response waveform (0.5–4.5 Hz) and the waveforms filtered into 1-Hz wide bands centered at 1 and 2 Hz, respectively. The response near 1 Hz is strongly modulated by attention but the response near 2 Hz is not. The onset of the 1st syllable of each sentence is marked by a dashed vertical line while the 3rd syllable is marked by a solid line. The shaded areas in this panel and panel **c** cover 1 SEM across participants (N = 15) on each side. The EEG signal is from channel Cz. **c** EEG and EOG response spectra. Spectral peaks at the sentential and phrasal rates are observed for EEG and EOG in both conditions. For each spectral peak, the P-value, mean, and SEM are shown in Supplementary Table 3. **d** The response phase extracted by the Fourier analysis. The shaded area covers the 95% confidence interval of the mean phase. If the phase difference between conditions is significant, stars are marked near the figure title. See text for the exact confidence interval. *P < 0.05; **P < 0.01

here may be a harmonic of the 1-Hz response rather than a separate response component.

Experiment 3 demonstrates that ocular activity is phase locked to the attended syllable within a sentence and a question naturally follows: Is ocular tracking of attended time moments a special phenomenon for speech processing or a general phenomenon for sequence processing? Two additional experiments are used to address this question, which employ non-linguistic auditory and visual sequences, respectively. In Experiment 4, the participants listen to an isochronous tone sequence and have to internally group every four tones into a perceptual group (Fig. 4a). Participants attend to the 1st tone in each perceptual group in one condition and attend to the 3rd tone in the other condition. The task is to detect if the attended tones are replaced by frequency modulated tones. The experiment further contrasts conditions in which the participants open or close their eyes. EOG signals are recorded in all conditions and eyetracking data are also collected when the eyes are open.

**Attention modulates ocular tracking of tone sequences**. In Experiment 4, when the eyes are open, a significant 1-Hz peak is observed in the response spectrum for vertical EOG, blink rate, and saccade rate measured by eyetracking (Fig. 4b). When the eyes are closed, significant 1-Hz spectral peaks are observed in vertical and horizontal EOG (Fig. 4e). In both the response waveform and in the phase of the 1-Hz response, it can be seen that the attentional focus, i.e., whether attention is oriented to the 1st or the 3rd tone, strongly influences the ocular responses. The response phase is extracted by applying the Fourier transform to the grand averaged response waveforms. When the eyes are open, at 1 Hz, the phase difference between conditions is 162° (127–194°/114–204°), 53° (−65° to 218°/−106° to 234°), 159° (115–197°/95–210°), −141° (−179° to −102°/−186° to −76°), and −106° (−182° to 30°/−233° to 77°), for the vertical EOG, horizontal EOG, blink rate, saccade rate, and pupil size, respectively. When the eyes are closed, the 1-Hz phase difference is 167° (77–265°/22–334°), and 147° (−52° to 278°/−53° to 301°) for vertical and horizontal EOG. Therefore, similar to Experiment 3, in Experiment 4 ocular activity is synchronized to the four-tone perceptual group and its phase is strongly modulated by temporal attention.

For tone sequences, it is flexible to define how many tones belong to each perceptual group, therefore we further test if similar results can be obtained when the participants group every eight tones into a perceptual group. In this experiment, referred to as Experiment 4b, ocular tracking of eight-tone perceptual groups is observed and its phase is also modulated by attention (Fig. 5). At 0.5 Hz, the phase difference between conditions is 190° (150–252°/133–288°), 172° (142–206°/133–221°), and 101° (52–145°/22–70°), for the blink rate, saccade rate, and pupil size, respectively. Therefore, ocular tracking of sequential structures is robust to sequence duration.

**Attention modulates ocular tracking of visual sequences**. Ocular synchronization to high-level structures in auditory sequences is shown in the previous four experiments. Ocular activity, however, directly affects vision rather than audition. To compare ocular tracking of auditory and visual sequences, the 5th experiment presents a sequence of visual shapes (200 ms stimulus onset asynchrony (SOA) between shapes) and the participants perform a temporal attention task similar to those used in Experiments 3 and 4. The visual sequence is periodic and each period contains 10 shapes, corresponding to 2 s in duration (Fig. 6a and Supplementary Fig. 3). The participants have to attend to either the 5th shape, a triangle, or the 10th shape, a circle, within each period. Eyetracking data demonstrate that blinks are synchronized to the 2-s stimulus period in the visual

sequence and the blink rate is low during the attended shape (Fig. 6). The saccade rate and pupil size also show significant tracking of the 2-s stimulus period. At 0.5 Hz, the phase difference between conditions is 183° (173–193°/175–191°), 162° (120–205°/101–218°), and 197° (100– 343°/53–384°), for the blink rate, saccade rate, and pupil size, respectively.

**Suppression of EOG power during stimulus**. Previous analyses all focus on how ocular activity tracks the internal structure of a sequence. By analyzing single-trial EOG power before, during, and after the sequence, it is observed that ocular activity is generally suppressed during the stimulus (Fig. 7). The EOG power gradually decreases after the stimulus onset and sharply increases after the stimulus offset.

**Ocular and neural tracking of non-isochronous speech**. Lastly, in all previous experiments, the syllables and sentences are presented at a constant rate. In Experiment 6, we test if the EOG/EEG response can still track spoken sentences when the SOA between syllables is jittered between 200 and 400 ms (Supplementary Fig. 4). The EOG and EEG responses are epoched based on the onset of each syllable, and each epoch is 1 s in duration starting from 250 ms before the syllable onset. The same baseline is removed from all epochs and the baseline is the mean value averaged over the pre-stimulus period (the 250-ms period before syllable onset) of all syllables. In each trial, the response to the first sentence is excluded from the analysis since it contains the transient response to sound onset. The response to each of the remaining 44 syllables from 11 sentences is averaged across trials, resulting in 44 event-related responses.

The waveform of the neural/ocular response evoked by each syllable is shown in Fig. 8a. A decoding analysis reveals that each syllable in a sentence evokes a different response waveform (Supplementary Fig. 4). To further compare the results in Experiment 6 with the frequency-domain results in previous experiments, we time-warp the neural responses to the temporally jittered stimuli to simulate the response to a 4-Hz isochronous syllable sequence (see Methods). The spectra of the time-warped responses are shown in Fig. 8b. A significant 1-Hz spectral peak is observed for EEG, vertical EOG, and horizontal EOG. This result demonstrates that the occurrence of neural and ocular tracking of sentences is not limited to isochronously presented stimuli.

## Discussion

Sensory and motor functions are often inseparable parts of natural behavior. For example, animals employ the sensory system to sense danger and use the motor system to escape. Similarly, speech perception and production are interleaved during natural conversations. The frequent interactions between sensory and motor systems demonstrate an effective interface between them, which provides the neuroanatomical basis to involve motoric mechanisms in perceptual tasks that do not require overt movements[2,4,9,43,44]. Here, a series of experiments demonstrate that, ocular muscle activity synchronizes to mentally constructed high-level sequence structures, which is potentially driven by temporal attention.

It is demonstrated across six experiments that EOG/blink activity can track high-level structures in sequences and is modulated by temporal attention. How the blink rate varies within the duration of a high-level structure is summarized in Fig. 9a. Stimulus-synchronized variation in the blink rate is stronger in the visual experiment compared with the auditory experiments. The time lag at which the blink rate reaches its maximum is illustrated in Fig. 9b. Since the sequences are periodic, the time lag of a response peak cannot be uniquely determined: Time lags differing by an integer number of periods are equivalent. Therefore, time lags shown in Fig. 9b are suggestive.

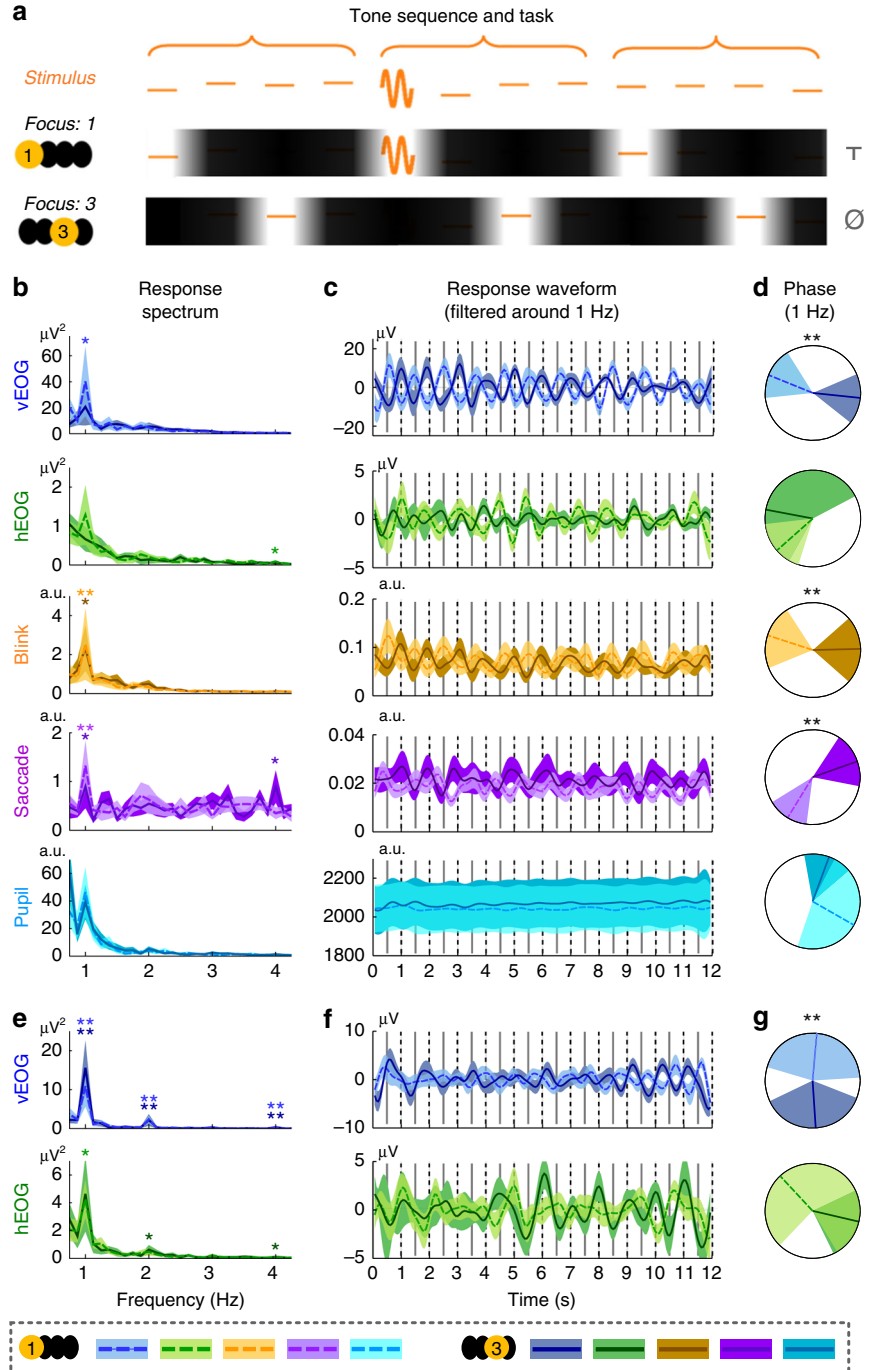

**Fig. 4** Ocular activity tracks high-level perceptual structures in a tone sequence and its phase is modulated by temporal attention. **a** Stimulus and task. The stimulus is an isochronous sequence of tones and the listeners are asked to group every four tones into a perceptual group. The task is to attend to the 1st syllable of every perceptual group in one condition and attend to the 3rd syllable in the other condition. Unattended syllables are shaded in the illustration. The listeners detect whether the attended tones are replaced by frequency modulated tones. In the illustration, T means that a target appears while Ø means that no target appears. Panels **b**–**d** show the results from the eyes open condition while panels **e**–**g** show the results from the eyes closed condition. In all these panels, dashed lines and lighter color indicate conditions attending to the 1st tone while solid lines and darker color indicate conditions attending to the 3rd tone. **b** and **e** Response spectrum. The shaded area in panels **b**, **c**, **e**, and **f** covers 1 SEM across participants ($N = 16$) on each side. The onset of the 1st tone of each perceptual group is marked by a dashed vertical line while the 3rd tone is marked by a solid line. The $P$-value, mean, and SEM of each spectral peak are shown in Supplementary Table 4. **c** and **f** Response waveform filtered around 1 Hz, which is strongly modulated by attention. **d** and **g** The response phase extracted based on the Fourier analysis. The shaded area covers the 95% confidence interval of the mean phase. If the phase difference between conditions is significant, stars are marked on top of the unit circle. *$P < 0.05$; **$P < 0.01$

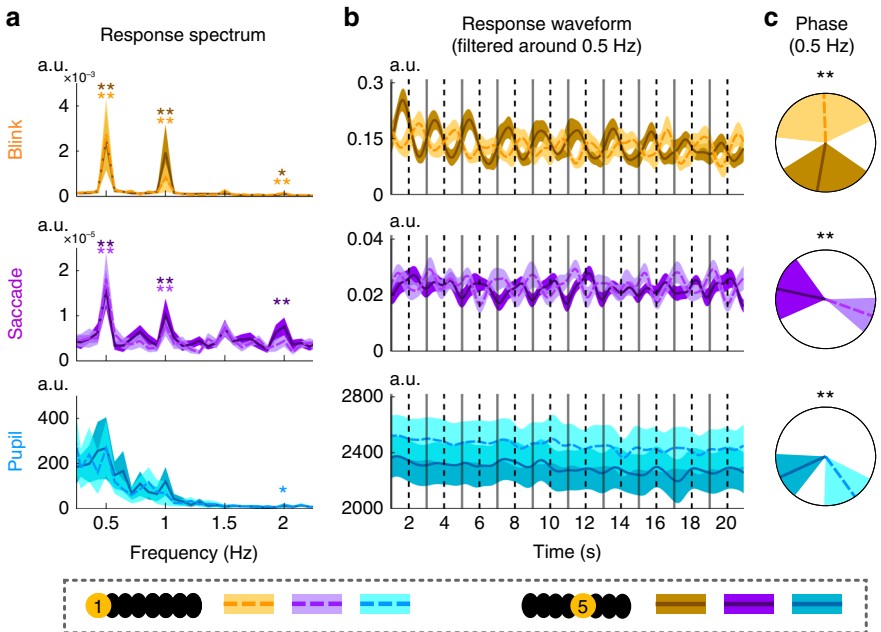

**Fig. 5** Ocular tracking of eight-tone perceptual groups in a tone sequence. **a** Response spectrum. A response peak at 0.5 Hz is observed for the blink/saccade rate (see Supplementary Table 5 for the *P*-value, mean, and SEM, *N* = 16), corresponding to the 2-s period of the eight-tone perceptual group. The figure setup is the same as Fig. 4. **b** Response waveform filtered around 0.5 Hz, which is strongly modulated by attention. **c** The response phase extracted by the Fourier analysis. The shaded area covers the 95% confidence interval of the mean phase. *P < 0.05; **P < 0.01

Blinks can only be detected when the eyes are open while EOG can monitor ocular activity whether the eyes are open or closed. To quantify the relationship between vertical EOG and blinks, we calculated the cross-correlation between vertical EOG and the binary blink signal in single trials for experiments that record using both EOG and eyetracking. It is observed that the two signals fluctuate in phase (Fig. 9c). The correlation coefficient at time lag 0 is $0.59 \pm 0.02$ (mean ± SEM across participants, 95% confidence interval: 0.56–0.62). Therefore, a blink is associated with a positive peak in vertical EOG. The timecourse of EOG and when it reaches its maximum within a high-level structure are also summarized in Fig. 9a, b. In Experiments 3–5, which range from speech listening experiments to visual sequence processing experiments, it can be seen that EOG/blink activity always tends to peak about 600–800 ms after the attended time moment. Based on this observation, it can be postulated that, when judging valid/invalid sentences in Experiment 2, blinks are triggered by the last syllable in a sentence, at which time the participants could be sure that what they hear is a valid sentence. Further experiments, however, are needed to validate whether the eyes generally blink ~600–800 ms after the attended time moment when processing any sensory sequence.

When the eyes are open, blinks recorded by eyetracking are strongly correlated with vertical EOG (Figs. 2 and 9c), suggesting that vertical EOG mostly reflects eyelid movements. In some experiments, structure-tracking activity is also found in the saccade rate and pupil size. Nevertheless, it remains possible that these structure-tracking saccadic and pupil responses are driven by blinks for the following reason: Blinks drive saccades and apparent changes in the pupil size, which can only be removed for blinks detected by eyetracking. For some blinks, however, the eyelid only partially closes. These eye blinks are not detected by eyetracking and therefore may affect the saccade and pupil size signals.

It is well established that cortical activity can track individual sound items in a sequence, e.g., syllables in speech[45] and notes in music[19]. Recently, it has also been shown that cortical activity can track higher-level structures, such as phrases and sentences in speech[20,46,47] and meters in music[19]. The current study

demonstrates that, on top of cortical activity, eye activity can also track high-level structures in speech and tone sequences. Neural and ocular tracking of high-level sequential structures, e.g., sentences and tone groups, cannot be explained by neural encoding of low-level acoustic/visual features, since no such features are correlated with the high-level structures. Instead, they must reflect either higher-level rule-based operations[48] or related modulatory processes, e.g., attention.

Attentional control constitutes an integral part of the mental process parsing sequential structures. The perception of musical meters has been attributed to rhythmic allocation of attention to the metrical accent[22]. Similarly, during speech processing, the amount of attention paid to each linguistic unit greatly varies. Specifically, at the word level, the initial syllable contributes more to word recognition than later syllables[49]. Neurophysiologically, the initial syllable in a word elicits a stronger neural response[50], which reflects a general increase in auditory sensitivity near the word onset[51]. At the sentence level, behavioral studies have shown that the brain preferentially processes words in the semantic focus during listening[25]. Therefore, temporal attention is naturally aligned to hierarchical sequential structures and potentially leads to ocular tracking of sequential structures, whether the task explicitly requires temporal attention or not. The current experiment has only tested simple sequential structures, e.g., short sentences and simple tone groups. Therefore, further experiments are needed to validate the general relationship between ocular activity and attentional focus for more complex speech and musical sequences.

Low-frequency neural activity has been proposed as an index for temporal attention[29,52]. Previous studies have demonstrated a clear relationship between top-down attention and low-frequency neural tracking of sound features. In a complex auditory scene consisting of multiple sound streams, a number of studies have observed stronger neural tracking of the attended sound stream[10,35]. Studies on relatively simple sensory stimuli have also indicated that the phase of ongoing neural activity can reflect spontaneous attentional fluctuations and can relate to behavior[53,54]. For sequences with high-level structures, using speech stimuli, it has been shown that the grouping of syllables

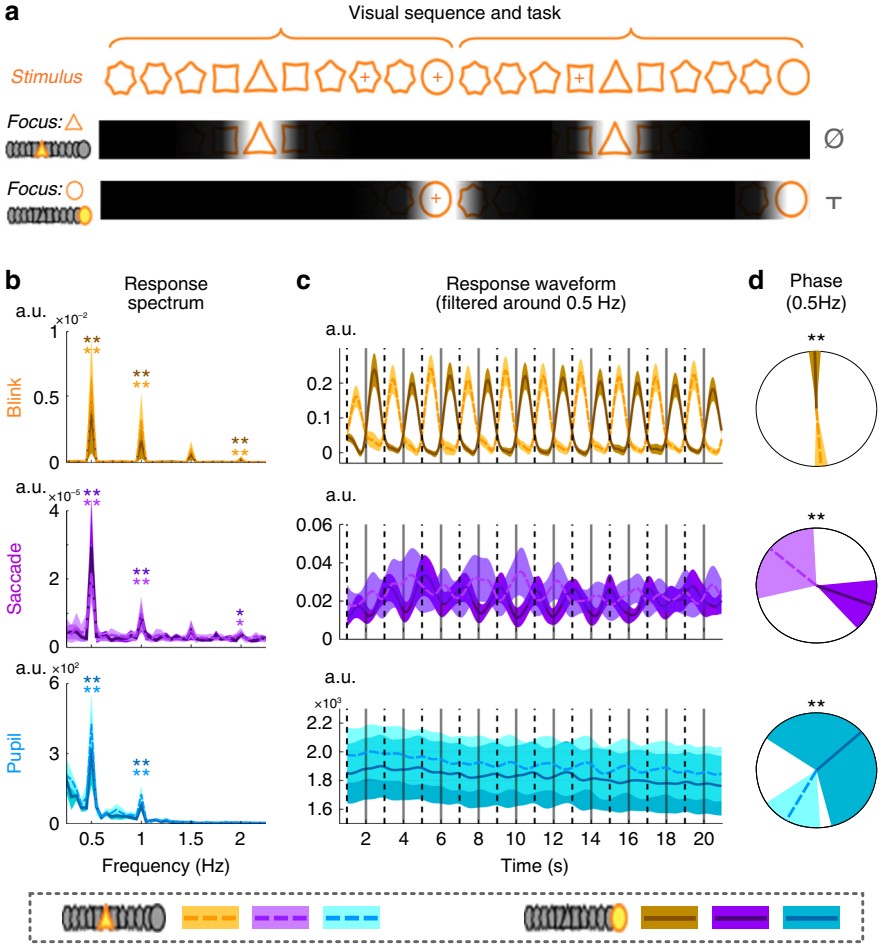

**Fig. 6** Ocular activity tracks structures in a visual sequence and its phase is modulated by temporal attention. **a** Stimulus and task. The stimulus is a sequence of shapes that repeats every 2 s. The task is to attend to the 5th shape in each 2-s period, i.e., a triangle, in one condition and to attend to the 10th shape, i.e., a circle, in the other condition. Unattended visual shapes are shaded in the illustration. The listeners detect whether a cross appears at the center of the attended shape. In the illustration, T means that a target appears while Ø means that no target appears. The shapes are described in details in Supplementary Fig. 3. **b** Response spectrum, in which response peaks at 0.5 Hz are observed (see Supplementary Table 6 for the P-value, mean, and SEM), corresponding to neural tracking of the 2-s period in the visual sequence. In all panels, dashed lines and lighter color indicate conditions attending to the 5th shape while solid lines and darker color indicate conditions attending to the 10th shape. The onset of the 5th shape is marked by a dashed vertical line while the 10th shape is marked by a solid line. In panels **b** and **c**, shaded area is 1 SEM across participants (N = 16) on each side. **c** Response waveform filtered around 0.5 Hz, which is strongly modulated by attention. The shaded area covers 1 SEM across participants on each side. **d** The response phase extracted by the Fourier analysis. The shaded area covers the 95% confidence interval of the mean phase. *P < 0.05; **P < 0.01

into multisyllabic words critically depends on attention[41]. The current study extends these previous studies by showing that the phase of low-frequency neural activity tracks attended syllables in a sentence and by showing ocular synchronization to the attentional focus.

Why is oculomotor activity synchronized to temporal attention? It is known that parts of the attention network[42,55], including the frontal eye fields (FEF) and posterior parietal cortex (PPC), are also involved in controlling eye movements and eye blinks[36,37]. Furthermore, the timing circuits in the brain also overlap with the motor system and includes, e.g., the supplementary motor cortex, parietal cortex, cerebellum, and basal ganglia[3]. It has also been hypothesized that sensory and motor systems are both entrained to the attended stimulus rhythm[29], and synchronized activity in distributed cortical networks provides a potential mechanism to coordinate distributive neural processing[56]. In sum, attentional control, interval timing, and motor control all involve distributed networks in the frontal/parietal lobes and sub-cortical nuclei, which potentially enables the interactions between temporal attention and ocular activity. The specific neural networks involved in ocular synchronization

to temporal attention or high-level sequential structures, however, could be highly complex and need to be characterized by future brain imaging studies. For example, previous work has shown that the cerebellum controls the timing of blinks during eyelid conditioning in which a tone precedes an aversive stimulus by a fixed time interval[57]. Nevertheless, it has been suggested that the cerebellum is specialized for absolute timing, e.g., estimation of the duration of a single time interval, while the basal ganglia is specialized for relative timing[58], e.g., timing relative to a steady beat, and therefore is more likely to be involved in the tasks in the current experiment.

Furthermore, pupil dilation and eyelid opening have long been used as indicators for attention/arousal level and processing difficulty[38–40]. It has been shown that, when listening to speech, a larger pupil size is observed for syntactically more complex sentences[59]. The current study, however, focuses on whether ocular activity can track the internal structure of a sequence, rather than whether it can reflect the overall arousal/attentional level. In terms of the temporal dynamics of eye/eyelid activity, previous studies have shown that the blink rate increases after finishing either an auditory or visual task[40,60,61]. Similarly, the current

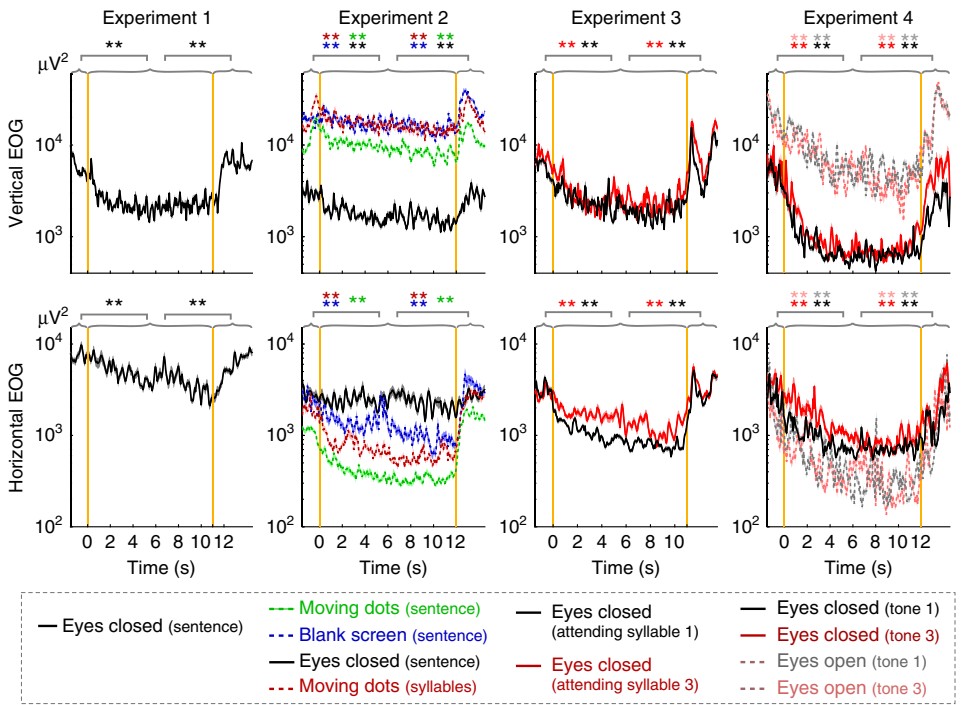

**Fig. 7** Timecourse of the EOG power before, during, and after the stimulus. The vertical orange lines denote the onset and offset of the stimulus sequence. The upper and lower panels show vertical and horizontal EOG, respectively. The curves show the average of the single-trial EOG power, i.e., square of the EOG waveform. The solid and dashed curves show the eyes-closed and eyes-open conditions, respectively. Shaded area is 1 SEM across participants on each side. The EOG power is lower during the stimulus, compared with the pre-stimulus and post-stimulus periods (bootstrap, significance shown by stars in the same color as the lines, *P < 0.05, **P < 0.01). See Supplementary Table 7 for the exact P-value, mean, and variance measures

study also finds a large increase in the blink rate after each sound/ visual sequence (Fig. 7). More importantly, here we find that even when the eyes are closed vertical EOG activity, which is likely to reflect eyelid movements, still increases after a sound sequence.

Ocular tracking of the internal structure of a sequence has been reported when watching a video story[62,63]: Fewer blinks are observed during visual scenes that are crucial to the story and visual scenes with fewer blinks are better remembered. The current study, however, demonstrate ocular tracking of the internal structure of auditory sequences, which have two critical differences from visual sequences. First, blinks block visual but not auditory information. Therefore, reducing the blink rate directly benefits visual but not auditory processing. Nevertheless, we still observe a low blink rate at the attended moments during auditory target detection tasks, consistent with findings in the visual experiments (Fig. 9). Second, in speech and many other natural sound, information comes rapidly. Here, in Experiments 3 and 4, the brain has to regularly switch between attended and unattended states within each second. Such rapid fluctuations in attention are rarely required during natural visual processing but common and beneficial for auditory scene analysis[64]. In sum, the current results show that the timing of blinks can be controlled with sub-second precision during active listening, whether the eyes are closed or open.

A syllabic/tone-rate ocular response is observed in some conditions but the magnitude is much smaller than that of the sentential/group-rate ocular response. The syllabic/tone-rate ocular response is potentially evoked by the auditory input. It is also possible, however, that it is a harmonic of the response tracking larger structures. Previous studies have shown that ocular responses to sound can be evoked in various conditions. For example, a loud or unexpected sound can evoke eye blinks and saccades as a part of the auditory startle/blink reflex[65] and the orientation reflex[1], respectively, even in completely dark environments. When sounds are played in a sequence, like in the

current study, each sound can still evoke a blink response[66]. The syllabic-rate ocular response observed in the current study can potentially be explained by these previously observed links between auditory processing and eye activity.

The auditory and motor systems are closely coupled, especially during speech communication. Previous research on the motor activation during speech listening has mostly focused on the cortical level[67,68]. The current study, however, directly observes speech-tracking responses in ocular muscle activity, demonstrating that auditory perception can induce overt muscle responses even outside the speech articulatory system. The sentence-tracking ocular response, however, must have a cortical origin since the sentential structure in the current experiment is defined by its syntactic structure instead of sensory cues, and syntactic analysis is implemented cortically. Peripheral muscle responses provide definitive evidence for the motor system's involvement in speech perception tasks but muscle activity is not a necessary outcome of cortical motor activation. In other words, even when ocular motor areas in cortex are activated, additional mechanisms could inhibit peripheral muscle responses. Therefore, it is highly likely that ocular muscle activity per se does not causally contribute to speech comprehension while the related motor cortical activation influences speech comprehension via mechanisms that will be discussed in the following.

A number of theories have been put forward about how motor cortical activity contributes to speech perception. First, motor activation may contribute to linguistic operations ranging from phonetic to syntactic processing[5,7,43] and indeed phoneme discrimination tasks have been causally related to the motor cortex[14,15,17]. It has also been proposed that motor cortical activation may be responsible for vocal learning and sensory–motor integration[4]. On top of these speech-specific mechanisms, the motor cortex may also play a more general role in directing spatial[42,44] and temporal attention[31,33,55,69]. Although this function has mostly been investigated using nonspeech stimuli, the

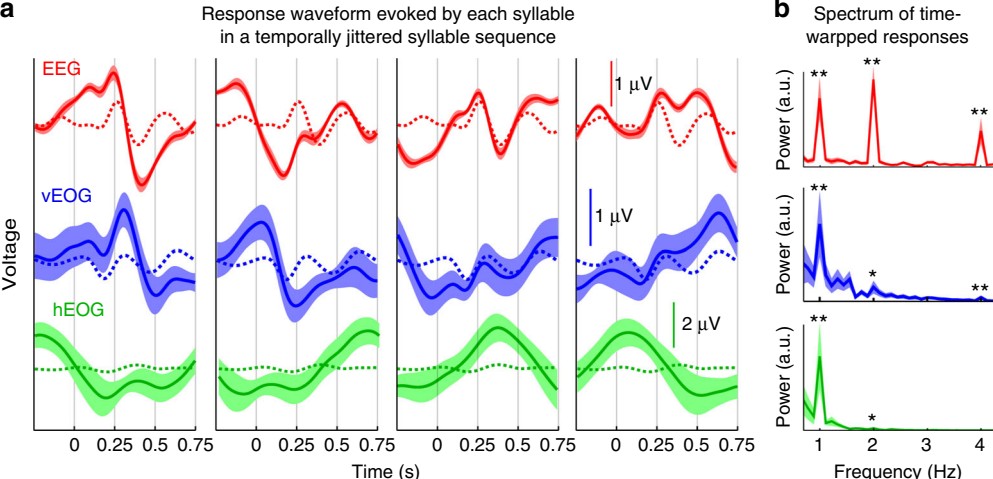

**Fig. 8** EOG and EEG tracking of sentences for temporally jittered syllables. **a** EEG and EOG responses to each of the four syllables in a sentence. The response is averaged over sentences and participants, and the shaded area covers 1 SEM. The dotted curve shows the response averaged over all four syllables. **b** The response to temporally jittered syllables is time warped to simulate the response to an isochronous syllable sequence. The spectrum of the simulated response shows a peak at the sentential rate for EEG, vertical EOG, and horizontal EOG. See Supplementary Table 8 for the exact $P$-value, mean, and variance measures. $*P < 0.05$; $**P < 0.01$

current results strongly suggest that temporal attention during speech comprehension also engages the motor system, providing another explanation for the activation of motor cortex during speech comprehension.

Additionally, oculomotor activity may also relate to speech processing as a mechanism for audiovisual integration. Speech can be understood without any visual cue but looking at the speaker's face can facilitate speech comprehension especially in noisy listening environments, since facial cues are temporally correlated with acoustic features of speech[70]. Furthermore, human listeners tend to look at the object related to the spoken words they hear, even without explicit instructions or when the object has disappeared when they hear the relevant word[71]. Furthermore, eye movement patterns can reflect grammatical forms when watching a blank screen when listening to a story[72]. In the current study, no visual input could facilitate speech perception, but it remains possible that the multisensory circuits are still reflexively engaged to actively absorb visual information based on the high-level speech structures.

In summary, we observe ocular tracking of mentally constructed high-level structures when listening to connected speech or other sound sequences, which provides definitive and easy-to-measure evidence for motor system's involvement in speech/auditory perception. It is proposed that cortical motor activation reflected in the structures-tracking ocular response is a general mechanism to allocate temporal attention during sequence processing.

## Methods

**Participants**. Totally, 119 participants took part in the study (19–29 years old, mean age, 22.2 years; 56 female). All participants were graduate or undergraduate students at Zhejiang University, with no self-reported hearing loss or neurological disorders. All participants were right-handed[73]. Experiments 1, 3, and 6 had 15 participants; Experiment 2 had 32 participants; Experiments 4, 4b, and 5 had 16 participants. Six participants took part in two experiments and no participant took part in three or more experiments. The experimental procedures were approved by the Institutional Review Board of the Zhejiang University Interdisciplinary Center for Social Sciences. The participants provided written consent and were paid.

**Speech stimuli**. Speech stimuli were presented in Experiments 1, 2, 3, and 6. The speech stimulus consisted of a sequence of independently synthesized syllables (neospeech synthesizer, the male voice, Liang). All syllables were adjusted to the same intensity, measured by the RMS, and same duration, i.e., 250 ms (see ref.[28] for details). The syllables were concatenated without introducing any additional

acoustic gap in between. All sentences used in this study ($N = 120$) had four syllables, with the first two syllables constructing a noun phrase and last two syllables constructing a verb phrase. In Experiments 1 and 2, 12 distinct sentences (48 syllables) were presented in a trial and therefore each trial was 12 s in duration. In Experiment 3, 11 distinct sentences were presented in a trial, before which a bisyllabic word was presented (see Experimental procedures and tasks). The two syllables in the word were independently synthesized and adjusted to 250 ms.

**Tone sequences**. Experiment 4 presented tone sequences. In each trial, 48 pure tones were isochronously presented. The SOA between tones was 250 ms. The frequency of each tone was randomly chosen within a two-semitone range centered at 1500 Hz (following a uniform distribution in logarithmic frequency). The tone duration was 50 ms, with 10 ms onset and offset cosine ramps. The participants were asked to perceptually group every four tones together and in the following every four tones was called a perceptual group. In each perceptual group, one of the four tones might be frequency modulated to create an outlier. The modulation rate was 40 Hz and the tone frequency was modulated between 100 Hz above and below the center frequency.

Experiment 4b also presented isochronous tone sequences. The tone sequence was created by inserting a tone after each tone in the sequence used in Experiment 4. The frequency of the inserted tone had the same distribution as the tones in the original sequence, i.e., uniform distribution within a two-semitone range centered at 1500 Hz. Consequently, the duration of each sequence was doubled and each sequence had 96 tones. Participants were asked to group every eight tones into a perceptual group.

**Visual stimuli**. In Experiment 2, the visual stimulus consisted of cyan (RGB: 0, 200, 250) dots moving in a black background (RGB: 0, 0, 0). On average, 136 dots appeared in a rectangular region (about 22° by 18° in the horizontal and vertical directions). The velocity of each dot was generated independently. It was the vector sum of a constant component $v$ and a time-varying component $u(t)$. The speed was ~7° per second for both the constant and time-varying component. The moving direction of $v$ was independently generated for each dot and the moving direction of $u(t)$ was independently generated for each dot and each time moment, both drawing from a uniform distribution between 0° and 360°. The position and velocity of each dot was updated at the screen's refresh rate (60 Hz).

In Experiment 5, a sequence of shape contours were displayed on the screen and all shapes were centered on the screen. The shape contours were white (RGB: 255, 255, 255) and the background was dark gray (RGB: 30, 30, 30). The shapes were shown in Supplementary Fig. 3 and the diameter of the circumscribed circle was about 1.8°. Each shape was displayed on the screen for 100 ms and a blank screen was shown for 100 ms until the next shape was displayed. In a sequence, 10 shapes constituted a period (2 s in duration) that repeated 11 times in each trial. Occasionally, a cross (RGB: 150, 150, 150) was displayed at the center of the screen during the presentation of a shape. In each stimulus period, at most two crosses were displayed.

**Experimental procedures and tasks**. In Experiments 1, 2, and 6, each stimulus consisted of a sequence of four-syllable sentences (Fig. 1a). Experiment 1 also

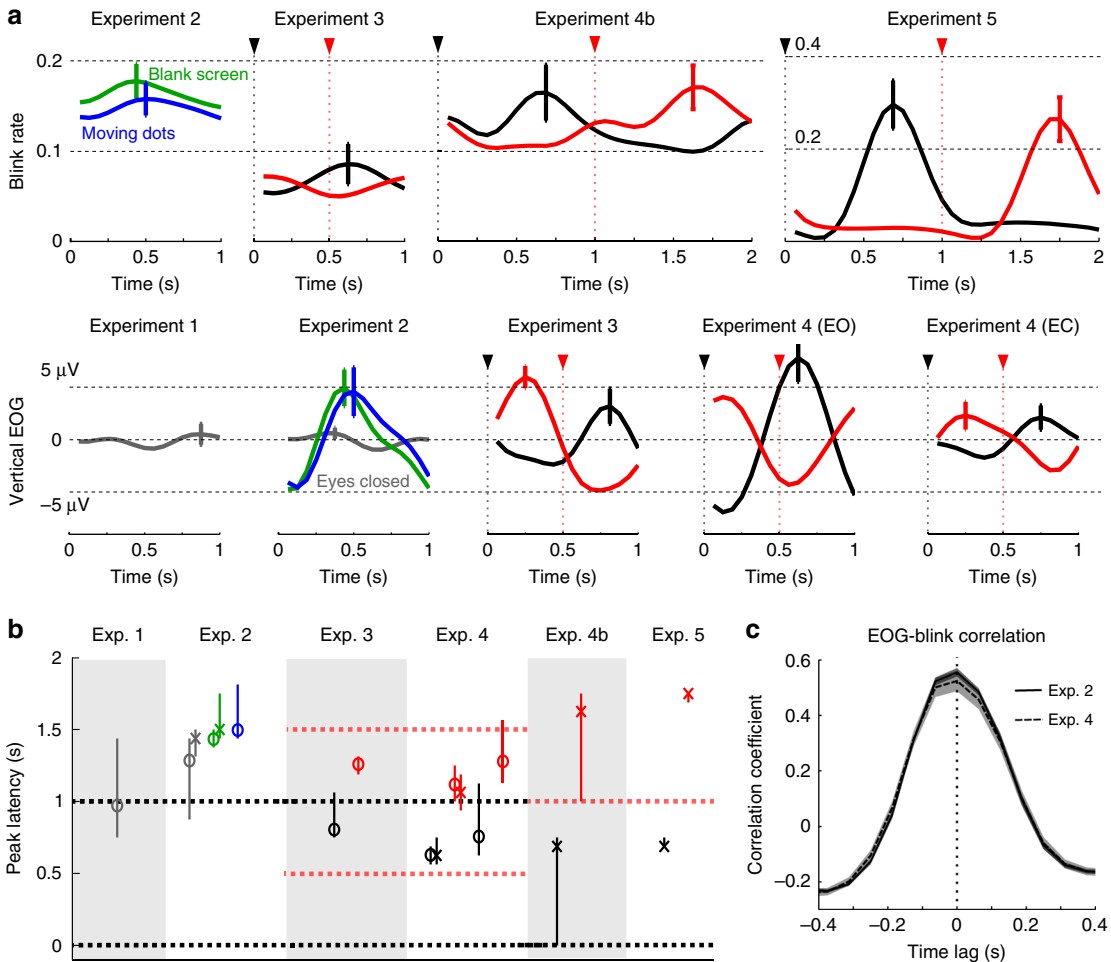

**Fig. 9** Waveform of the blink rate and vertical EOG within each high-level structure in Experiments 1–5. **a** Timecourses of the blink rate and vertical EOG within each high-level structure. For Experiments 3–5, the onset of the attended unit is denoted by a triangle on the top. Black color represents conditions when attention is directed to the beginning of the structure while red color represents conditions when attention is directed to the midpoint of the structure. Other conditions are labeled in the figure. In Experiment 4, when the eyes are closed, clear phase opposition between conditions is only observed near the end of each trial (Fig. 4f) and therefore only the last 3 s of responses in each trial are averaged in this analysis. In Experiment 5, the response is aligned based on the 5th shape, which therefore is at time 0. **b** The time lag at which the blink/EOG signal reaches its maximal. Since the sequence structure is periodic, the estimated time lag could be advanced or delayed by an integer number of sequence periods. The error bar shows the 95% confidence interval, evaluated using bootstrap. In the bootstrap procedure, time lags from resampled data are converted into phase angles and averaged using the circular mean. **c** The cross-correlation function between single-trial blink signal and vertical EOG. A positive correlation is observed at time lag 0 in Experiments 2 and 4, in which EOG and eyetracking are simultaneously recorded

contained four other experimental conditions using speech materials with different linguistic structures (pairs of nouns) and the results would be reported separately.

Experiment 1: Thirty trials were presented. In 50% trials, the last two syllables of the sentence sequence were removed so that the last sentence only consisted of a two-syllable noun phrase. After each trial, the participants pressed different keys to indicate whether all sentences were complete or one of them was truncated. The participants made a correct response in 97 ± 1% trials (mean ± SEM across participants). After the key press, the next trial was presented after a silent interval randomized between 1 and 2 s (uniform distribution). Before the experiment, the participants were familiarized with one normal trial and one outlier trial that were not used in the experiment.

Experiment 2: The experiment was divided into five blocks. Each block consisted of 35 trials and five of them were outlier trials that were not included in the EEG or eyetracking analysis. Blocks 1–4 presented sentence materials while block 5 presented random syllables. Blocks 1 and 2 were identical, which construct the main condition in the experiment. In the first 2 blocks and also block 5, the participants freely viewed random dots moving on a monitor. In blocks 3 and 4, the monitor and light were turned off. The participants kept their eyes open in block 3 and closed in block 4. Each block contained 35 trials. Block 1 was always presented first to investigate if the ocular response adapted over time. The order of blocks 2–5 was randomized.

In blocks 1–3, normal trials contained 12 sentences while the outlier trials were identical to the normal trials except that the four syllables of a randomly chosen sentence were shuffled. The shuffled syllables did not construct any meaningful

expression. In block 5, a normal trial presented 48 random syllables while in an outlier trial four consecutive syllables at a random position were replaced by a sentence. At the beginning of each trial, instructions were given about what kind of trials were normal or outlier trials. After each trial, the participants pressed different keys to indicate outlier trials or normal trials. After the key press, the next trial was presented after a silent interval randomized between 1 and 2 s (uniform distribution). Before the experiment, participants listened to two sample trials to get familiar with the tasks.

Experiment 3: Each trial began with a bisyllabic word that was randomly chosen from six candidates, i.e., /dà dì/, /fēng fù/, /gǎi gé/, / háo huá/, / kāng kǎi/, and /yīn yuè/, with equal probability. 1 s after the bisyllabic word, a 11-s sentence sequence was presented (Supplementary Fig. 2). The bisyllabic word consisted of two syllables that shared the same initial consonant and the shared initial consonant was defined as the target consonant. The experiment was divided into two blocks. In one block, the participants were asked to attend to the 1st syllable in every sentence and detect if the target consonant was the initial consonant of any attended syllable. In the other block, the participants did the same consonant detection task but attended to the 3rd syllable of every sentence. The order of the two blocks was counterbalanced across participants. Each block contained 60 trials.

In each sentence sequence, one and only one sentence contained a syllable starting with the target consonant. That syllable, however, might appear as the 1st, 2nd, 3rd, or 4th syllable of that sentence with equal probability (i.e., 15 trials for each position). When the target consonant appeared at the attended position (the 1st syllable in one block and the 3rd syllable in the other block), the participant

should press a key as soon as possible. A correct response was recorded only if the key press fell in a 1750-ms window starting from the onset of the target consonant. If the target consonant appeared an unattended position, the participants should not press any key throughout the trial. 1 s after the sequence offset, auditory feedback ("right" or "wrong" in Chinese) was given. After the auditory feedback, the next trial was played after another key press. The participants' responses were correct in 95 ± 1% and 93 ± 1% trials (mean ± SEM across participants) in the blocks attending to the 1st and 3rd syllables, respectively.

Before the main experiment, participants went through a training session to get familiar with the task. The training session was identical to the main experiment except that it was terminated after the participants made 9 correct responses in 10 consecutive trials. For all participants, the training session ended after 13 ± 1 and 18 ± 4 trials (mean ± SEM across participants) in the blocks attending to the 1st and 3rd syllables, respectively.

Experiment 4: A sequence of tones were presented in each trial and occasionally a tone was replaced by a frequency-modulated outlier. The experiment divided into four blocks. In two blocks, the participants were instructed to attend to the 1st tone in each perceptual group and detect how many times the attended tone was replaced by an outlier, i.e., a frequency modulated tone. The other two blocks employed the same task but the participants attended to the 3rd tone instead of the 1st tone in each perceptual group. The first three perceptual groups in each trial did not contain any outlier and in these three perceptual groups the tone at the attended position was cued by amplifying the tone by a factor of 4.

Each block had 40 trials. In 26 trials, no outliers appeared and only these 26 trials were used for the EOG and eyetracking analyses. In the other 14 trials, half of them had two outliers and the other half had one outlier. For the two blocks attending to the same tone in a perceptual group, the participants closed their eyes in one block and opened their eyes in the other block. When the participants opened their eyes they were instructed to look at a blank screen in front of them. In each block, the total number of outliers was the same at each position of a perceptual group. At the end of each trial, the participants had to report whether 0, 1, or 2 attended tones were replaced by outliers by pressing 0, 1, or 2 on a keyboard. After the key press, auditory feedback was provided, i.e., "right" or "wrong" in Chinese, and the next trial was presented 1–2 s after the auditory feedback. When the eyes were open, the participants made correct responses in 93 ± 1% and 90 ± 2% trials when attending to the 1st and 3rd tones, respectively. When the eyes were closed, the performance was 93 ± 2% and 92 ± 2% for the 1st and 3rd tones, respectively (mean ± SEM across participants).

Before the main experiment, two behavioral screening sections was applied. In the first section, participants had to detect whether the 1st tone in a perceptual group was replaced by an outlier, while in the second section the participants detected whether the 3rd tone was replaced. The screening section was the same as the main experiment except that when the participants made a wrong response they could choose to repeat the stimulus. In each block of the screening section, 12 trials were presented and if the participants made 9 or more correct responses they passed the training section. Otherwise, another block was presented. At most three blocks were presented in each screening section. Thirty-three out of 69 participants passed the screening section and were invited for the main experiments of either Experiment 4 or Experiment 4b described in the following.

Experiment 4b: Experiment 4b only differed from Experiment 4 in two aspects: First, since each perceptual group consisted of eight tones, the participants attended to the 1st tone in one block and attended to the 5th tone in the other block. Second, only two blocks were presented and the participants kept their eyes open. The participants made correct responses in 93 ± 2% and 92 ± 2% trials when attending to the 1st and 5th tones, respectively (mean ± SEM across participants).

Experiment 5: The participants viewed a sequence of shapes and occasionally a cross might appear at the center of a shape. The participants were asked to selectively attend to one shape and detect whether a cross appeared during that shape 0, 1, or 2 times. They responded by pressing 0, 1, or 2 on a keyboard at the end of each trial. After the key press, visual feedback was provided on the screen, i.e., "right" or "wrong" in Chinese. The next visual sequence was presented 1–2 s after the visual feedback. The experiment divided into two blocks. The attended shape was triangle in one block and circle in the other block. The order of the two blocks was counterbalanced across participants. Each block had 40 trials. In a trial, no cross was displayed during the first or last stimulus period. Across all trials, the same number of crosses were displayed at each of the 10 shapes within a stimulus period.

In 26 trials, no crosses appeared and only these 26 trials were used for the eyetracking analysis. In the other 14 trials, seven trials had two outliers. The participants made correct responses in 91 ± 2% and 85 ± 1% trials when attending to the triangle and circle, respectively (mean ± SEM across participants). A training section was given before the main experiment for each condition, in which 12 trials were presented. The participants passed the training section if they made nine or more correct responses. Otherwise they had to repeat the training section. Sixteen out of 21 participants passed the training section within three attempts and participated in the main experiment.

Experiment 6: The trial structure and the task were the same as those in blocks 1–4 of Experiment 2. However, the SOA between syllables was jittered between 200 and 400 ms, following a uniform distribution. One hundred and five trials (including 90 normal trials and 15 outlier trials) were presented and the participants had a break after every 35 trials. The order of trials was randomized. The participants had their eyes closed in the whole experiment.

**Recordings.** EOG was recorded in all experiments that consisted of eyes closed conditions, i.e., all experiments except for Experiments 4b and 5. EEG was recorded in Experiments 1–3 and 6. Eyetracking was recorded in Experiments 2, 4, 4b, and 5. EEG and EOG were recorded using a Biosemi ActiveTwo system. In Experiments 1, 3, and 6, 64 EEG electrodes were recorded (10–20 electrode system), while in Experiment 2 only five EEG electrodes were recorded (Cz, Fz, FCz, FC3, and FC4). To record EOG, two electrodes were placed at the left and right temples and their difference was the horizontal EOG (right minus left). Another two electrodes were placed above and below the right eye and their difference was the vertical EOG (upper minus lower). Two additional electrodes were placed at the left and right mastoids and their average was the reference for EEG. The EEG/EOG recordings were low-pass filtered below 400 Hz and sampled at 2048 Hz. The EEG recordings were referenced to the average mastoid recording off-line and high-pass filtered above 0.7 Hz using a linear-phase finite impulse response (FIR) filter. To remove EOG artifacts in EEG, the horizontal and vertical EOG were regressed out using the least-squares method[41]. Occasional large artifacts in EEG/EOG, i.e., samples with magnitude > 1 mV, were set to zero.

In Experiments 2, 4, 4b, and 5, eyetracking data were recorded using a combined pupil and corneal reflection eye tracker at 500-Hz sampling rate (Eyelink Portable Duo, SR Research, Mississauga, Ontario, Canada). Participants were seated 58 cm from a monitor with their chin resting on a chinrest. At the beginning of each experimental block, a 9-point (3 × 3 square) calibration and validation was applied. In Experiment 2, after every 10 trials, a white dot appeared at the center of the screen in a black background for recalibration purposes. The participant had to fixate at the dot for more than 1 s and the experimenter had to confirm that the fixation was detected properly before continuing the experiment. In Experiments 4, 4b, and 5, a 9-point calibration and validation was applied after every 20 trials. Eyetracking data were only recorded during the stimulus.

In the eyetracking data, time intervals when the pupil could not be detected were defined as blinks. Time intervals satisfying the following three criteria were defined as saccades: motion > 0.1°, velocity > 30°/s, acceleration > 8000°/s². The pupil area was used to measure the pupil size and its unit was not calibrated. Blinks were associated with saccades and an apparent change in the pupil size. To isolate saccades not related to blinks, saccades surrounding a blink were removed from the analysis. Similarly, the pupil size data within a 200 ms interval after each blink was removed from the analysis since the pupil size apparently increases during eyelid opening.

After preprocessing, EOG and eyetracking data were all downsampled to 16 Hz, since the current study only focused on low-frequency responses below 4 Hz. When filtering data around a specific frequency (e.g., 0.5, 1, or 2 Hz), a linear-phase Hamming-window FIR filter was used (impulse response duration: 2 s) and the delay of the filter was compensated by shifting the data back by 1 s, which was the group delay of the filter. Unlike EEG/EOG measures that could not reliably measure the direct current (DC) component, the DC component in eyetracking data, e.g., blink/saccade rate and pupil size, was meaningful. Therefore, when filtering eyetracking measures, the DC component was removed before the filtering process and added back after filtering. In other words, filtering the eyetracking data did not change the mean blink/saccade rate or pupil size averaged over time.

**Frequency-domain analysis.** In the frequency-domain analysis, to avoid onset/offset effects and focus on steady-state neural activity, neural/ocular activity during the first and last second of each trial was not analyzed. Consequently, the analysis window was 10 s in Experiments 1 and 2, 9 s in Experiment 3, 8 s in Experiment 4, and 20 s in Experiments 4b and 5. The EEG/ocular responses in the analysis window were transformed into the frequency domain using the discrete Fourier transform (DFT) without any additional smoothing window. When averaging the response power over participants or electrodes, the geometric mean was used since the response power varied across participants and the arithmetic mean was sometimes dominated by a couple of participants. The geometric mean is the same as the arithmetic mean of individual power in a dB scale. Since the blink and saccade rates were very low in some participants, to avoid biasing the geometric mean by participants with a very low blink/saccade rate, a small value was added to the DFT power of individual participants before the group average. The small value was one thousandth of the root mean square value of the DFT power across all frequencies and participants. The circular mean was used to average the response phase[74].

In Experiment 6, the SOA between syllables was jittered so that the responses to hierarchical linguistic structures were not frequency tagged. The response to the temporally jittered syllables, however, could be time warped to simulate the response to syllables presented at a constant rate using the procedure described in the Supplementary Methods. The time-warped response was used for frequency-domain analysis, following the same procedure used for other experiments.

**Statistical tests.** All tests were based on bias-corrected and accelerated bootstrap[75]. In the bootstrap procedure, all the participants were resampled with replacement 10,000 times.

Spectral peak: The statistical significance of a spectral peak at frequency $f$ was tested by comparing the response power at $f$ with the power averaged over two neighboring frequency bins just below $f$ using bootstrap. We did not consider the neighboring frequency bins above $f$ since their power was generally weaker than the

power at $f$ due to the 1/f trend in background neural activity. Therefore, the significance test used here was relatively conservative. In the bootstrap procedure, the power spectrum was averaged over a group of resampled participants. The test is one-sided: If the response is stronger at the target frequency compared with neighboring frequencies in $A$% of the resampled data, the significance level is $(100A + 1)/10001$. For experiments using sentence stimuli, significance tests were only applied to the responses at 1, 2, and 4 Hz, corresponding to the sentential, phrasal, and syllabic rates, and a FDR correction was applied to these three frequencies. To be consistent, in Experiment 4, the same three frequencies were used for significance tests, even though 2 Hz did not correspond to any structure. Similarly, in Experiments 4b and 5, since the high-level structures repeated at 0.5 Hz, significance tests were applied at 0.5, 1, and 2 Hz.

Response phase: The $A$% confidence interval of the mean phase is the smallest angle that can cover $A$% of the 10,000 resampled mean phase. If the confidence interval does not include 0°, the response phase significantly deviates from 0 ($P = 1 − A$%). This test is two-sided.

Bootstrap is also used to estimate the SEM across participants and for this purpose participants were resampled with replacement 100 times.

## Data availability
Data and the MATLAB scripts to generate the figures are included in Supplementary Data 1.

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

## Acknowledgements

We thank David Poeppel, Jonathan Simon, and Huan Luo for helpful comments on earlier versions of this manuscript, Jipeng Duan and Naifei Su for assistance with the experiments, Zhuowen Shen and Yuhan Lu for editing the manuscript. Work supported by National Natural Science Foundation of China 31771248 (N.D.), 31500873 (N.D.), Zhejiang Provincial Natural Science Foundation of China LR16C090002 (N.D.), and research funding from the State Key Laboratory of Industrial Control Technology, Zhejiang University (N.D.).

## Author contributions

N.D. conceived the study. P.J., J.Z., and T.Z. implemented the experiments. P.J. performed Experiments 1 and 6; P.J. and T.Z. performed the Experiments 2 and 3; P.J. and J. Z. performed Experiments 4, 4b and 5. P.J. and N.D. analyzed the EEG data; P.J., J.Z., and N.D. analyzed the EOG data; T.Z., P.J., and N.D. analyzed the eyetracking data. N.D. wrote the manuscript. All authors edited the manuscript.

## Additional information

**Competing interests:** The authors declare no competing interests.

