## [Peer Review File · Nature Communications]

Reviewer #1 (Remarks to the Author):

Active tracking of high level speech structures in sensorimotor systems

This is an interesting study pioneering a relatively novel direction of research: How does active listening affect the way the eyes screen the visual environment?

The authors address this based on a unique stimulus material, which induces phase- and sentence-like structures without specific acoustic cues. While subjects listen out for specific syllables the authors measure behavioural performance, brain activity using EEG, and eye movements. Consistent with previous studies they find that dynamic brain activity aligns to the syllabic, phrasal and sentence structure on each respective time scale. The new observation here is that blinks of the eye dynamically align to the speech material on the sentence level as well. Given that the sentence structure could only be extracted based on linguistic, but not acoustic, information, the authors conclude that high-level language processes are involved in aligning ocular activity with the external sensory environment. Overall this demonstrates that higher cognitive processes affect the manner in which active sensing across the modalities gathers new sensory information.

Two main issues came to my mind when reading this article: Is this actually surprising, and is the authors interpretation as clear as they suggest?

Novelty and impact: Ocular activity in general is known to reflect task strategies and sensory saliency as shown by many studies on visual perception. Studies on decision making show a clear impact of self-refractory processes such as confidence in one's own decisions on pupil dilation. And finally, eye blinks have been shown to align to time periods of reduced attentional load, or to the overall temporal structure of a task design in a cognitive neuroscience study (e.g. inter-stimulus intervals; <https://www.ncbi.nlm.nih.gov/pubmed/7870552>; <https://www.ncbi.nlm.nih.gov/pubmed/19640888>; <https://www.ncbi.nlm.nih.gov/pubmed/6701241> ; <http://journals.plos.org/plosone/article?id=10.1371/journal.pone.0141242>

). With that in mind it seems little surprising that in the present study blinks align to some of the structure of the stimulus material. Blinks are necessary, so the only parameters the brain can adjust are how often and when to blink. As the authors show, this can be guided by sentence structure or the instruction to focus on a specific syllable. With the above studies in mind, this seems maybe a bit less surprising than the authors claim. I think the authors need to better integrate their results with the field of eye movement studies in vision and cognitive psychology. This would also help to clarify the actual novelty of the results presented here.

Interpretation with regards to language: the discussion remains ambiguous as to the precise mechanisms controlling ocular activity. At some point the authors conclude that these could be 'either higher-level linguistic operations or related modulatory processes such as attention'. In my view the case for an alignment of temporal is good, as demonstrated by Exp3. However, whether linguistic operations are necessary or not remains in my view unclear. In particular, I would not be surprised if the very same results were obtained with a similarly structured series of visual images, if these contained some multi-level hierarchical organization that would allow the brain to imply a specific temporal pattern to the series of images. In other words, do the authors believe that the reflection of temporal attention by eye blinks is specific to speech or acoustic stimuli, or is it rather a very generic process in which the brain organizes a temporal series of incoming information by some hierarchical rule in order to automatically determine when to pay more, and when to pay less attention? Given the need to blink (see above) this may be a process that is continuously guiding active sensing in general and across the modalities. One possibility to demonstrate the generic nature of this process would be a visual version of the experiment. All in all I find that the interpretation of the data in the context of language is too narrow and fails to acknowledge the generic principle behind the mechanisms that in this experiment align eye blinks to specific syllables.

In the following I present comments on some of the central results and arguments:

Introduction / I 58 ff: I find that the first question set out here has been addressed in some recent work (e.g. the Keitel paper cited; also Park & Gross PlosBiol 2017). The second part of the questions is unclear; what are different properties? Do you refer to function, or neural correlates? Also, the emphasis here is on motor networks, which from the preceding text would seem to refer to motor structures in the brain. But the data presented here focus on muscle activity in the eye, hence motor periphery. There seems to be a mismatch between the expectations set out in the introduction and the actual data.

Confounds of EOG and EEG / Figure 1: The presented analyses convince that the EOG signals are not confounded by brain activity. Anyway, the usual concern is the opposite, that scalp recordings of brain activity are confounded by ocular activity. One additional analysis that the authors could implement is to correlate EOG and EEG across trials within a participant. The between-subject correlation in Fig. 1D may be confounded by between subject variations in the relative quality of EOG or EEG recordings, but a within subject correlation would avoid this issue.

Specificity to blinks / Fig S1: I think this is a rather important result, which should be presented in the main text. The authors show that the EOG signal reflects blinks, not eye movements. Critical questions here are how saccadic eye movement were defined (velocity criteria etc). The manuscript remains very opaque to this analysis. Do these only contain macro- or also micro-saccades? I think this section requires more methodological information.

Specificity to sentence structure: this key result is buried in an experiment for which no data is shown and which is mentioned anecdotally, despite being central to the authors arguments. Why? I definitely would like to see the data for this control presented in the same way as for Exps 1-3 in Figure 2. The lack of a reporting of effect sizes makes it impossible to judge the quality of this central conclusion.

Mechanisms: on lines 212ff the authors set up two hypotheses as to the mechanisms underlying the tracking of speech in the EOG signal: Temporal attention and syntactic analysis. However, the study implements only a manipulation of temporal attention and hence can only assert the importance of this, but the data can't be used to arbitrate between the two hypotheses. I agree that the overall design of the stimuli probably necessitates the involvement of lexical analysis to extract sentence level structure, but this would allow ruling in or out a contribution of syntactic analysis.

Further comments:

The title is somewhat overstating the results, which in the end reduce to temporal attention marked by blinks of the eye.

The introduction prepares a big playing field, drawing particular importance of rhythmic brain activity and its role in coordinating brain networks. However, the study and its direct interpretation do not speak on brain activity at all and the reader's expectations towards rhythmic brain activity are not met. I would deemphasize this part in the introduction.

L 16: the citations with regards to neural synchrony and behaviour seem a bit outdated and there would be more suitable recent reviews (e.g. by the Siegel group, Schroeder & Co, or the Fries group).

L 31ff: This sentence is partly a tautology and does not connect well to the preceding text about speech and word selection.

L 183: Here the authors conclude that 'these analysis show that the listeners blink AT THE SAME POSITION OF A SENTENCE'. However, so far we have been presented only analyses of power, which ignore phase and hence position information.

L 260: Here the authors report a 90% confidence interval. Is 90 indeed correct? If so I wonder whether this is sufficient tight to allow clear conclusions.

L 827: The citation Hyojin et al should be Park et al. (the last name is Park)

Statistics: Overall I found the statistics to be appropriate. However, it would be good to report precise p-values and measures of effect size, where appropriate. Even when using bootstrapping, the authors could e.g. report the differences in EOG power (or whatever measure is under consideration). As it stands, I find that the statistical tests are somewhat minimally described.

Reviewer #2 (Remarks to the Author):

The manuscript describes a series of EEG and EOG experiments, during spoken sentence comprehension, that reveal rhythmic movements of the eyes and eyelids corresponding to the same frequencies at which the EEG signal shows peaks in power. Especially at the ends of sentences, but also at phrasal breaks and syllable/word breaks, the vertical component of the EOG indicates that eye muscles and eyelid muscles are substantially activated. An attention-manipulation condition suggests that the rhythmic motor movement of vEOG and hEOG may be more something of a mix of domain-general attentional processes and domain-specific linguistic processes. The results also extend to non-rhythmic delivery of the sentences as well. The findings confirm an active role for sensorimotor systems in otherwise-passive comprehension of spoken language. The experimental design is sound, the control conditions are appropriate, and the results are compelling. The findings make a transformative contribution to the field that will be cited frequently, and I strongly support publication of this manuscript, with only minor suggested revisions.

Minor Comments:

For future follow-up work, I encourage the authors to further explore the domain-general attentional component of this result by trying out a condition that uses musical notes in a similar nested structure. The sentence/melody could be four notes long, one phrase could have two notes from one octave while the other phrase has two notes from a different octave. Will the results look the same?

In the discussion section on Motor Activation During Speech Perception, the second paragraph gets incredibly close to hinting at the Motor Theory of Speech Perception (which has been criticized but revised versions fit quite a bit of data). In this paragraph, it might be worth briefly acknowledging this theoretical proposal for the role of the motor system in speech comprehension (e.g., Liberman & Mattingly, 1985), its updated framework (e.g., Galantucci, Fowler, & Turvey, 2006), and supporting evidence as well (e.g., Fadiga, Craighero, Buccino, Rizzolatti, 2002).

In the paragraph that follows that one, the authors mention Huettig et al.'s (2011) results where participants move their eyes toward a blank region of the screen that used to contain an object that is being referred to by a noun in the speech stream. To accompany that finding, it might be worth also mentioning the work of Huette, Winter, Matlock, Ardell, and Spivey (2014), where they found that a rather abstract grammatical component of sentences (i.e., verbal aspect) also altered the pattern of eye movements on a blank screen.

Minor Corrections:

- line 6, change “where the sound comes,” to “where the sound comes from,”
- line 24, change “presented a rapid rate” to “presented at a rapid rate”
- line 31, change “Studies on sentence processing has proposed” to “Studies on sentence processing have proposed”
- line 77 change “may serve as synchronization signal” to “may serve as a synchronization signal”
- line 78 change “coordinate neural processing massive cortical networks” to “coordinate neural processing across massive cortical networks”
- line 398 change “The current study extend” to “The current study extends”
- Several of the references are missing volume numbers or page numbers.

Reviewer #1 (Remarks to the Author):

Active tracking of high level speech structures in sensorimotor systems

This is an interesting study pioneering a relatively novel direction of research: How does active listening affect the way the eyes screen the visual environment?

The authors address this based on a unique stimulus material, which induces phase- and sentence-like structures without specific acoustic cues. While subjects listen out for specific syllables the authors measure behavioural performance, brain activity using EEG, and eye movements. Consistent with previous studies they find that dynamic brain activity aligns to the syllabic, phrasal and sentence structure on each respective time scale. The new observation here is that blinks of the eye dynamically align to the speech material on the sentence level as well. Given that the sentence structure could only be extracted based on linguistic, but not acoustic, information, the authors conclude that high-level language processes are involved in aligning ocular activity with the external sensory environment. Overall this demonstrates that higher cognitive processes affect the manner in which active sensing across the modalities gathers new sensory information.

We thank the reviewer for the accurate summary and positive evaluation. In the revised manuscript, we have added 3 new experiments to demonstrate that ocular activity can also track high-level temporal structures in non-linguistic visual or auditory sequences. The new experiments also show that fewer blinks are recorded at the attended versus unattended time intervals, regardless of the sensory modality, suggesting a common sensorimotor mechanism to control temporal attention with sub-second precision.

Two main issues came to my mind when reading this article: Is this actually surprising, and is the authors interpretation as clear as they suggest?

In the following, we briefly summarize our responses to these two major concerns. We have also modified our interpretations and emphasized the novelty of the results in the manuscript.

Interpretation: Our previous interpretation was indeed too narrow. Following the constructive suggestions by the reviewers, we have conducted 3 new experiments and additional analyses to reach a more general conclusion: Ocular activity can actively track task-related temporal structures in sensory sequences. Furthermore, it is shown that blinks (and also EOG activity recorded from the closed eyes) are suppressed at attended time moments during auditory/speech processing, similar to what is observed during visual processing (Fig. 9), suggesting a domain-general mechanism for sequence structure parsing and temporal attention.

Novelty: The current study extends two lines of studies. On the one hand, recent studies show that neural activity in traditional language areas tracks high-level linguistic structures such as phrases and sentences (Pallier et al., 2011; Ding et al., 2016; Sheng et al., 2018). The current study, however, show that peripheral ocular muscle activity can also track sentential structures. This result demonstrates that sensorimotor circuits, including the oculomotor circuits, are involved in processing high-level sequential structures. Furthermore, it suggests that temporal attention may explain why ocular activity is synchronized to sequential structures.

On the other hand, previous studies have demonstrated the blink rate increases *after*, e.g., auditory discrimination tasks, and can synchronize to dynamic visual scenes. The current study, however, show that ocular activity can track sequence structures *during* speech/sound listening. Furthermore, the sequence structures in this study are mentally constructed based on either syntactic knowledge (Experiments 1, 2, 3, and 6) or experimental instructions (Experiments 4 and 4b), rather than physically defined by sensory cues at structural boundaries. More importantly, we show that eye blinks can capture *rapid, i.e., sub-second, fluctuations in temporal attention*. Rapid switching between attended and unattended states on the order of a few hundred milliseconds is rarely needed for natural visual processing but can greatly benefit speech perception in a noisy environment: The brain selectively attends to time intervals when speech power exceeds noise power and ignores other time intervals, a strategy known as glimpsing or dip listening (Miller and Licklider, 1950; Cooke, 2006).

Furthermore, we show that even *when the eyes are closed*, vertical EOG still tracks the temporal structure of the auditory input. Eyelid closure blocks the potential visual input and removes the physiological needs to blink, e.g., to lubricate the eyeball. Therefore, the current results strongly demonstrate that ocular tracking of temporal structures is an intrinsic property of the brain.

Novelty and impact: Ocular activity in general is known to reflect task strategies and sensory saliency as shown by many studies on visual perception. Studies on decision making show a clear impact of self-refectory processes such as confidence in one's own decisions on pupil dilation. And finally, eye blinks have been shown to align to time periods of reduced attentional load, or to the overall temporal structure of a task design in a cognitive neuroscience study (e.g. inter-stimulus intervals; <https://www.ncbi.nlm.nih.gov/pubmed/7870552>; <https://www.ncbi.nlm.nih.gov/pubmed/19640888>; <https://www.ncbi.nlm.nih.gov/pubmed/6701241>; <http://journals.plos.org/plosone/article?id=10.1371/journal.pone.0141242>).

With that in mind it seems little surprising that in the present study blinks align to some of the structure of the stimulus material. Blinks are necessary, so the only parameters the brain can adjust are how often and when to blink. As the authors show, this can be guided by sentence structure or the instruction to focus on a specific syllable. With the above studies in mind, this seems maybe a bit less surprising than the authors claim. I think the authors need to better integrate their results with the field of eye movement studies in vision and cognitive psychology. This would also help to clarify the actual novelty of the results presented here.

Thank you for pointing out these very relevant studies. We have now extended the discussion on the relationship between ocular activity and cognitive tasks.

"...pupil dilation and eyelid opening have long been used as indicators for attention/arousal level and processing difficulty (Beatty, 1982; Stern et al., 1984; McGinley et al., 2015). It has been shown that, when listening to speech, a larger pupil size is observed for syntactically more complex sentences (Schluroff, 1982). The current study, however, focuses on whether ocular activity can track the internal structure of a sequence, rather than whether it can reflect the overall arousal/attentional level. In terms of the temporal dynamics of eye/eyelid activity, previous studies have shown that the blink rate increases after finishing either an auditory or visual task (Stern et al., 1984; Fukuda, 1994; Siegle et al., 2008). Similarly, the current study also finds a large increase in the blink rate after each sound/visual

sequence (Fig. 7). More importantly, here we find that even when the eyes are closed vertical EOG activity, which is likely to reflect eyelid movements, increases after a sound sequence.

Ocular tracking of the internal structure of a sequence has been reported when watching a video story (Nakano et al., 2009; Shin et al., 2015): Fewer blinks are observed during visual scenes that are crucial to the story and visual scenes with fewer blinks are better remembered. The current study, however, demonstrate ocular tracking of the internal structure of auditory sequences, which have 2 critical differences from visual sequences. First, blinks block visual but not auditory information. Therefore, reducing the blink rate directly benefits visual but not auditory processing. Nevertheless, we still observe a low blink rate at the attended moments during auditory target detection tasks, consistent with findings in the visual experiments (Fig. 9). When the eyes are closed, the phase relationship between the blink rate and attended time moments shows higher individual variability (Fig. 4) and appears to be more consistent for speech than tone sequences (Figs. 3 and 4). Second, in speech and many other natural sound, information comes rapidly. Here, in Experiments 3 and 4, the brain has to switch from attended and unattended states within each second. Such a rapid switch is rare during visual processing but common and beneficial for auditory scene analysis (Miller et al., 1951; Cooke, 2006). In sum, the current results show that the timing of eye blinks can be controlled with sub-second precision during active listening, whether the eyes are closed or not.”

Interpretation with regards to language: the discussion remains ambiguous as to the precise mechanisms controlling ocular activity. At some point the authors conclude that these could be 'either higher-level linguistic operations or related modulatory processes such as attention'. In my view the case for an alignment of temporal attention is good, as demonstrated by Exp3. However, whether linguistic operations are necessary or not remains in my view unclear. In particular, I would not be surprised if the very same results were obtained with a similarly structured series of visual images, if these contained some multi-level hierarchical organization that would allow the brain to imply a specific temporal pattern to the series of images. In other words, do the authors believe that the reflection

of temporal attention by eye blinks is specific to speech or acoustic stimuli, or is it rather a very generic process in which the brain organizes a temporal series of incoming information by some hierarchical rule in order to automatically determine when to pay more, and when to pay less attention? Given the need to blink (see above) this may be a process that is continuously guiding active sensing in general and across the modalities. One possibility to demonstrate the generic nature of this process would be a visual version of the experiment. All in all I find that the interpretation of the data in the context of language is too narrow and fails to acknowledge the generic principle behind the mechanisms that in this experiment align eye blinks to specific syllables.

Thanks for the very valuable suggestion. We have now conducted a new experiment using visual sequences, i.e., Experiment 5. In the experiment, 10 visual shapes are presented isochronously and repetitively. The participants have to attend to one shape based on the instruction and detect if a cross appears inside the shape. The results show that blinks are strongly coupled to the visual sequence and the blink rate peaks about 700 ms after the target shape.

Critically, we find that the alignment between blinks and temporal attention is similar for speech, non-speech auditory sequences, and visual sequences (Fig. 9), suggesting a common mechanism during sequence processing.

“In Experiments 3-5, which range from speech listening experiments to visual sequence processing experiments, it can be seen that EOG/blink activity always tends to peak about 600-800 ms after the attended time moment. Based on this observation, it can be postulated that, when judging valid/invalid sentences in Experiment 2, blinks are triggered by the last syllable in a sentence, at which time the participants could be sure that what they hear is a valid sentence. Further experiments, however, are needed to validate whether the eyes generally blink ~600-800 ms after the attended time moment when processing any sensory sequence.”

The new Fig. 6 and Fig. 9 are shown in the following.

Figure 6. Ocular activity tracks structures in a visual sequence and its phase is modulated by temporal attention. A) Stimulus and task. The stimulus is a sequence of shapes that repeats every 2 s. The task is to attend to the 5th shape in each 2-s period, i.e., a triangle, in one condition and to attend to the 10th shape, i.e., a circle, in the other condition. Unattended visual shapes are shaded in the illustration. The listeners detect whether a cross appears together with the attended shape. The shapes are described in details in Fig. S3. B) Response spectrum, in which response peaks at 0.5 Hz are observed (see Table 6 for the P-value, mean, and SEM), corresponding to neural tracking of the 2-s period in the visual sequence. In all panels, dashed lines and lighter color indicate conditions attending to the 5th shape while solid lines and darker color indicate conditions attending to the 10th shape. The onset of the 5th shape is marked by a dashed vertical line while the 10th shape is marked by a solid line. In panels B and C, shaded area is 1 SEM on each side. C) Response waveform filtered around 0.5 Hz, which is strongly modulated by attention. The shaded areas cover 1 SEM across participants on each side. D) The response phase extracted by the Fourier analysis. The shaded areas cover the 95% confidence interval of the mean phase. * $P < 0.05$, ** $P < 0.01$

Figure 9. Summary of blink/EOG activity within a high-level structure. A) Timecourses of blinks and vertical EOG within the high-level structure in each experiment. For Experiments 3-5, the onset of the attended units is denoted by a triangle. Black lines indicate conditions when attention is directed to the beginning of the structure while red lines indicate conditions when attention is directed to the midpoint of the structure. In Experiment 4, when the eyes are closed, a clear phase opposition between conditions is only observed near the end of each trial (Fig. 4F) and therefore only the last 3 s of responses were averaged. In Experiment 5, the response is aligned based on the 5th shape, which therefore is at time 0. B) The time lag at which the blink/EOG signal reaches its maximal. Since the sequence structure is periodic, the estimated time lag could be advanced or delayed by an integer number of sequence periods. The error bar shows the 95% confidence interval, evaluated using bootstrap. In the bootstrap procedure, time lags from resampled data are converted into phase angles and averaged using the circular mean. C) The cross-correlation function between single-trial blink signal and vertical EOG. A positive correlation is observed at time lag 0 in Experiments 2 and 4, in which EOG and eyetracking are simultaneously recorded.

In the following I present comments on some of the central results and arguments:

Introduction / 1 58 ff: I find that the first question set out here has been addressed in some recent work (e.g. the Keitel paper cited; also Park & Gross PlosBiol 2017). The second part of the questions is unclear; what are different properties? Do you refer to function, or neural correlates? Also, the emphasis here is on motor networks, which from the preceding text would seem to refer to motor structures in the brain. But the data presented here focus on muscle activity in the eye, hence motor periphery. There seems to be a mismatch between the expectations set out in the introduction and the actual data.

Thanks for pointing out this issue and we have thoroughly modified the Introduction to set up a clear expectation for the current study. First, we now discuss how peripheral muscle activity can facilitate active sensing and may be modulated by speech processing:

“... Furthermore, transcranial magnetic stimulation (TMS) of the motor cortex can alter how the brain processes auditory syllables or words (Meister et al., 2007; Möttönen et al., 2012), and during TMS speech processing can modulate peripheral tongue/lip muscle activity (Fadiga et al., 2002; Watkins et al., 2003)....

... Studies on sensory processing have proposed that selective information processing in time, i.e., temporal attention, is implemented by low-frequency neural oscillations in the sensorimotor system (Jones et al., 2002; Large and Palmer, 2002; Atteveldt et al., 2014; Nobre and van Ede, 2018), and can be facilitated by overt movements (Phillips-Silver and Trainor, 2005; Schroeder et al., 2010; Morillon and Baillet, 2017)....”

Second, we clearly mention muscle activity when setting up the questions:

“... It is less clear, however, whether processing high-level structures in speech and other sounds engages the motor system and whether motor activity, either cortical or muscle activity, spontaneously tracks such larger structures without any movement-related task.

... Within a single sequence, e.g., a speech stream, it remains to be established whether the phase of sensorimotor activity is locked to the units that are preferentially processed, and whether such activity can modulate muscle activity.”

Confounds of EOG and EEG / Figure 1: The presented analyses convince that the EOG signals are not confounded by brain activity. Anyway, the usual concern is the opposite, that scalp recordings of brain activity are confounded by ocular activity. One additional analysis that the authors could implement is to correlate EOG and EEG across trials within a participant. The between-subject correlation in Fig. 1D may be confounded by between subject variations in the relative quality of EOG or EEG recordings, but a within subject correlation would avoid this issue.

Thanks for the useful suggestion. We have now analyzed the correlation between EOG and EEG in single trials and the results are shown in Fig. S1 for each participant. The 1-Hz EOG power is not significantly correlated with the 1-Hz EEG power.

Specificity to blinks / Fig S1: I think this is a rather important result, which should be presented in the main text. The authors show that the EOG signal reflects blinks, not eye movements. Critical questions here are how saccadic eye movement were defined (velocity criteria etc). The manuscript remains very opaque to this analysis. Do these only contain macro- or also micro-saccades? I think this section requires more methodological information.

We have moved the previous Fig. S1 to the main manuscript and show all ocular measures for the new Experiments (see the Figs. 2, 4, 5, and 6). We have also reported how saccades are defined (i.e., motion > 0.1°, velocity > 30°/s, acceleration > 8000°/s², which are the default parameter in the Eyelink system). Saccades defined this way are dominated by macrosaccades.

Specificity to sentence structure: this key result is buried in an experiment for which no data is shown and which is mentioned anecdotally, despite being central to the authors arguments. Why? I definitely would like to see the data for this control presented in the same way as for Exps 1-3 in Figure 2. The lack of a reporting of effect sizes makes it impossible to judge the quality of this central conclusion.

We have now added the control condition to Fig. 2. The mean and SEM of the 1-Hz peak are now reported in Table 2 for each condition.

Mechanisms: on lines 212ff the authors set up two hypotheses as to the mechanisms underlying the tracking of speech in the EOG signal: Temporal attention and syntactic analysis. However, the study implements only a manipulation of temporal attention and hence can only assert the importance of this, but the data can't be used to arbitrate between the two hypotheses. I agree that the overall design of the stimuli probably necessitates the involvement of lexical analysis to extract sentence level structure, but this would allow ruling in or out a contribution of syntactic analysis.

We have rewritten the introduction to Experiment 3. Now the only aim of the experiment is to test if temporal attention modulates the phase of ocular activity.

Further comments:

The title is somewhat overstating the results, which in the end reduce to temporal attention marked by blinks of the eye.

We have changed the previous title, "*Active Tracking of High-level Speech Structures in Sensorimotor Systems: Evidence from the Brain and the Eyes*", to "*Active Tracking of Task-relevant Structures in Eye Activity during Speech and Auditory Sequence Perception*", since the new experiments show that the phenomenon applies to non-linguistic materials. We have also removed the word "sensorimotor systems" and focus more on the behavior measure, i.e., "eye activity".

We did not directly use the word "blink", since the eyes were closed in several experiments and calling vertical EOG activity blinks may be confusing to some readers. We did not directly use the word "attention" either, since the tasks in Experiments 1 and 2 do not explicitly ask the listeners to attend to specific words. In the discussion, it is mentioned that the listeners may naturally pay different amounts of attention to different words during speech listening but this conclusion needs further experiments to validate.

The introduction prepares a big playing field, drawing particular importance of rhythmic brain activity and its role in coordinating brain networks. However, the study and its direct interpretation do not speak on brain activity at all and the reader's expectations towards

rhythmic brain activity are not met. I would deemphasize this part in the introduction.

Thanks for the helpful suggestions and we have rewritten most parts of the Introduction to deemphasize discussions on neural activity.

L 16: the citations with regards to neural synchrony and behaviour seem a bit outdated and there would be more suitable recent reviews (e.g. by the Siegel group, Schroeder & Co, or the Fries group).

We have updated the literature and the current references include:

Siegel M, Buschman TJ, Miller EK. Cortical information flow during flexible sensorimotor decisions. Science 2015, 348(6241): 1352-1355.

Fries P. Rhythms for cognition: communication through coherence. Neuron 2015, 88(1): 220-235.

Atteveldt Nv, Murray MM, Thut G, Schroeder CE. Multisensory Integration: Flexible Use of General Operations. Neuron 2014, 81: 1240-1253.

Since we have now deemphasized the discussion on neural synchrony, these studies are referred to in the Discussion.

L 31ff: This sentence is partly a tautology and does not connect well to the preceding text about speech and word selection.

Indeed. This sentence is now removed.

L 183: Here the authors conclude that 'these analysis show that the listeners blink AT THE SAME POSITION OF A SENTENCE'. However, so far we have been presented only analyses of power, which ignore phase and hence position information.

We have removed this sentence.

L 260: Here the authors report a 90% confidence interval. Is 90 indeed correct? If so I wonder whether this is sufficient tight to allow clear conclusions.

We have now reported both the 95% and 99% confidence interval.

L 827: The citation Hyojin et al should be Park et al.
(the last name is Park)

We have corrected the name.

Statistics: Overall I found the statistics to be appropriate. However, it would be good to report precise p-values and measures of effect size, where appropriate. Even when using bootstrapping, the authors could e.g. report the differences in EOG power (or whatever measure is under consideration). As it stands, I find that the statistical tests are somewhat minimally described.

We have now listed the mean and SEM of each response power measure in Tables 1-8. The 95% and 99% confidence interval for phase differences are reported in the Results.

References

- Cooke M (2006) A glimpsing model of speech perception in noise. *Journal of the Acoustical Society of America*
- Ding N, Melloni L, Zhang H, Tian X, Poeppel D (2016) Cortical tracking of hierarchical linguistic structures in connected speech. *Nature Neuroscience*
- Miller GA, Licklider JCR (1950) The Intelligibility of Interrupted Speech. *Journal of the Acoustical Society of America*
- Pallier C, Devauchelle A-D, Dehaene S (2011) Cortical representation of the constituent structure of sentences. *PNAS*
- Sheng J et al., (2018) The Cortical Maps of Syntactic Hierarchical Linguistic Structures during Speech Perception. *Cerebral Cortex*.

Reviewer #2 (Remarks to the Author):

The manuscript describes a series of EEG and EOG experiments, during spoken sentence comprehension, that reveal rhythmic movements of the eyes and eyelids corresponding to the same frequencies at which the EEG signal shows peaks in power. Especially at the ends of sentences, but also at phrasal breaks and syllable/word breaks, the vertical component of the EOG indicates that eye muscles and eyelid muscles are substantially activated. An attention-manipulation condition suggests that the rhythmic motor movement of vEOG and hEOG may be more something of a mix of domain-general attentional

processes and domain-specific linguistic processes. The results also extend to non-rhythmic delivery of the sentences as well. The findings confirm an active role for sensorimotor systems in otherwise-passive comprehension of spoken language. The experimental design is sound, the control conditions are appropriate, and the results are compelling. The findings make a transformative contribution to the field that will be cited frequently, and I strongly support publication of this manuscript, with only minor suggested revisions.

We thank the reviewer for the accurate summary and very positive evaluation.

Minor Comments:

For future follow-up work, I encourage the authors to further explore the domain-general attentional component of this result by trying out a condition that uses musical notes in a similar nested structure. The sentence/melody could be four notes long, one phrase could have two notes from one octave while the other phrase has two notes from a different octave. Will the results look the same?

Thanks for the very valuable suggestion. We have now added new experiments to show that ocular activity can synchronize to mentally constructed temporal structures in non-speech auditory sequence. This experiment, and also the new visual experiment suggested by reviewer #1, has greatly extended our conclusion, generalizing the previous results on speech perception to general sequence processing.

Since musical melodies are generally complex and usually have acoustic cues related to the musical structures, we choose to use simple tone sequences in the experiments. We presented an isochronous tone sequence and asked the participants to perceptually group every 4 tones or 8 tones into a chunk. It was found that ocular activity can entrain to the nonlinguistic 4-tone or 8-tone chunk (Figs. 4 and 5). Furthermore, in different blocks, we asked the participants to pay attention to different notes within a chunk and showed that the phase of entrained activity is modulated by attention. Figures 4 and 5 are shown in the following. The two figures correspond to Experiments 4 and 4b, which use very similar stimuli but the participants have to group either 4 or 8 tones into a perceptual group.

Figure 4. Ocular activity tracks high-level perceptual structures in a tone sequence and its phase is modulated by temporal attention. A) Stimulus and task. The stimulus is an isochronous sequence of tones and the listeners are asked to group every 4 tones into a perceptual group. The task is to attend to the 1st syllable of every perceptual group in one condition and attend to the 3rd syllable in the other condition. Unattended syllables are shaded in the

illustration. The listeners detect whether the attended tones are replaced by frequency modulated tones. Panels B-D show the results from the eyes open condition while panels E-G show the results from the eyes closed condition. In all these panels, dashed lines and lighter color indicate conditions attending to the 1st tone while solid lines and darker color indicate conditions attending to the 3rd tone. B/E) Response spectrum. The shaded area in panels B, C, E, and F covers 1 SEM on each side. The onset of the 1st tone of each perceptual group is marked by a dashed vertical line while the 3rd tone is marked by a solid line. The P-value, mean, and SEM of each spectral peak are shown in Table 4. C/F) Response waveform filtered around 1 Hz, which is strongly modulated by attention. D/G) The response phase extracted based on the Fourier analysis. The shaded areas cover the 95% confidence interval of the mean phase. If the phase difference between conditions is significant, stars are marked one top of the unit circle. * $P < 0.05$, ** $P < 0.01$

Figure 5. Ocular tracking of 8-tone perceptual groups in a tone sequence. A) Response spectrum. A response peak at 0.5 Hz is observed for the blink/saccade rate (see Table 5 for the P-value, mean, and SEM), corresponding to the 2-s period of 8-tone groups. The figure setup is the same as Fig. 4. B) Response waveform filtered around 0.5 Hz, which is strongly modulated by attention. C) The response phase extracted by the Fourier analysis. The shaded areas cover the 95% confidence interval of the mean phase. * $P < 0.05$, ** $P < 0.01$

In the discussion section on Motor Activation During Speech Perception, the second paragraph gets incredibly close to hinting at the Motor Theory of Speech Perception (which has been criticized but revised versions fit quite a bit of data). In this paragraph, it might be worth briefly acknowledging this theoretical proposal for the role of the motor system in speech comprehension (e.g., Liberman & Mattingly, 1985), its updated framework (e.g., Galantucci, Fowler, & Turvey, 2006), and supporting evidence as well (e.g., Fadiga, Craighero, Buccino, Rizzolatti, 2002).

The motor theory of speech perception is indeed very relevant to the current study. We have now added relevant discussions in both *Introduction* and *Discussion*.

“Even when no overt movement is involved, it has also been proposed that sensorimotor mechanisms play critical roles in speech and auditory perception. It has been hypothesized that the motor cortex contributes to decoding phonetic information in speech (Liberman and Mattingly, 1985; Galantucci et al., 2006). Neurophysiological evidence consistent with this hypothesis has shown that neural activity can track acoustic features not only in auditory cortex (Nourski et al., 2009; Giraud and Poeppel, 2012) but also in broad frontal/parietal areas that overlap with the motor and attention networks (Chen et al., 2008; Zion Golumbic et al., 2013b; Cheung et al., 2016; Park et al., 2016; Wöstmann et al., 2016; Assaneo and Poeppel, 2018; Keitel et al., 2018). Furthermore, transcranial magnetic stimulation (TMS) of the motor cortex can alter how the brain processes auditory syllables or words (Meister et al., 2007; Möttönen et al., 2012), and during TMS speech processing can modulate peripheral tongue/lip muscle activity (Fadiga et al., 2002; Watkins et al., 2003).”

In the paragraph that follows that one, the authors mention Huettig et al.'s (2011) results where participants move their eyes toward a blank region of the screen that used to contain an object that is being referred to by a noun in the speech stream. To accompany that finding, it might be worth also mentioning the work

of Huettenlocher, Winter, Matlock, Ardell, and Spivey (2014), where they found that a rather abstract grammatical component of sentences (i.e., verbal aspect) also altered the pattern of eye movements on a blank screen.

Thanks for pointing out this interesting and relevant study. We have added it to the discussion.

“eye movement patterns can reflect grammatical forms when watching a blank screen when listening to a story (Huettenlocher et al., 2014).”

Minor Corrections:

- line 6, change “where the sound comes,” to “where the sound comes from,”

Corrected.

- line 24, change “presented a rapid rate” to “presented at a rapid rate”

Corrected.

- line 31, change “Studies on sentence processing has proposed” to “Studies on sentence processing have proposed”

Corrected.

- line 77 change “may serve as synchronization signal” to “may serve as a synchronization signal”

Corrected.

- line 78 change “coordinate neural processing massive cortical networks” to “coordinate neural processing across massive cortical networks”

Corrected.

- line 398 change "The current study extend" to "The current study extends"

Corrected.

- Several of the references are missing volume numbers or page numbers.

We have carefully checked the references and the missing volume/page numbers are added.

REVIEWERS' COMMENTS:

Reviewer #1 (Remarks to the Author):

The authors have addressed my concerns. The newly added experiments and the revised interpretation together make for a cohesive and clear story.

Reviewer #2 (Remarks to the Author):

The authors have revised the manuscript with careful attention to the detailed concerns raised by myself and the other reviewer. With several new experiments added, not merely verifying but greatly extending the main thesis, the new manuscript is greatly improved. The designs, analyses, and results for the new experiments are sound and compelling.

I strongly recommend publication. This article will be an oft-cited classic for this journal.